ecology, taxonomy and systematics, evolution

shipworms, teredinidae, macrobioerosion, rock-boring, freshwater ecology, palaeontology

**Author for correspondence:**
Daniel L. Distel
e-mail: d.distel@northeastern.edu

### PUBLISHING

# A rock-boring and rock-ingesting freshwater bivalve (shipworm) from the Philippines

J. Reuben Shipway[1], Marvin A. Altamia[1], Gary Rosenberg[2], Gisela P. Concepcion[3], Margo G. Haygood[4] and Daniel L. Distel[1]

[1]Ocean Genome Legacy Center, Department of Marine and Environmental Science, Northeastern University, Nahant, MA, USA
[2]Academy of Natural Sciences, Drexel University, Philadelphia, PA, USA
[3]Marine Science Institute, University of the Philippines, Diliman, Quezon City, Philippines
[4]Department of Medicinal Chemistry, University of Utah, Salt Lake City, UT, USA

JRS, 0000-0002-6838-1917; GR, 0000-0002-2558-7640; MGH, 0000-0001-5277-9544; DLD, 0000-0002-3860-194X

Shipworms are a group of wood-boring and wood-feeding bivalves of extraordinary economic, ecological and historical importance. Known in the literature since the fourth century BC, shipworms are both destructive pests and critical providers of ecosystem services. All previously described shipworms are obligate wood-borers, completing all or part of their life cycle in wood and most are thought to use wood as a primary source of nutrition. Here, we report and describe a new anatomically and morphologically divergent species of shipworm that bores in carbonate limestone rather than in woody substrates and lacks adaptations associated with wood-boring and wood digestion. The species is highly unusual in that it bores by ingesting rock and is among the very few known freshwater rock-boring macrobioeroders. The calcareous burrow linings of this species resemble fossil borings normally associated with bivalve bioerosion of wood substrates (ichnospecies *Teredolites longissimus*) in marginal and fully marine settings. The occurrence of this newly recognized shipworm in a lithic substrate has implications for teredinid phylogeny and evolution, and interpreting palaeoenvironmental conditions based on fossil bioerosion features.

## 1. Background

Teredinids, commonly referred to as shipworms, are a group of predominantly marine, obligate wood-boring (xylotrepetic) and wood-feeding (xylotrophic) bivalves. As the primary consumers of woody materials across the world's oceans, shipworms play a critical role in liberating and cycling the recalcitrant carbon sequestered in lignocellulosic material, from shallow coastal driftwoods to deep-sea wood falls and mangrove stilt roots, seagrass rhizomes and coconut seeds [1,2]. Yet this prolific ability to degrade wood causes billions of dollars in damage to coastal structures, seagoing vessels and fishing equipment per annum, and as a result, shipworms are considered economically important pest organisms [1].

This unusual xylotrepetic and xylotrophic lifestyle has led to several unique anatomical specializations. In contrast to most bivalves, the valves are greatly reduced in size and located at the extreme anterior end of the body, covering little of the visceral mass and offering limited protection to the animal. Instead, the shell features tiny denticulated ridges across its surface, and functions as a drilling tool for burrowing in wood [3]. Upon ingestion, wood particles are typically stored and digested in the caecum, a sac-like extension of the stomach that is characteristic of the family [2]. In lieu of protective valves, the shipworm's vermiform body is covered by a calcareous

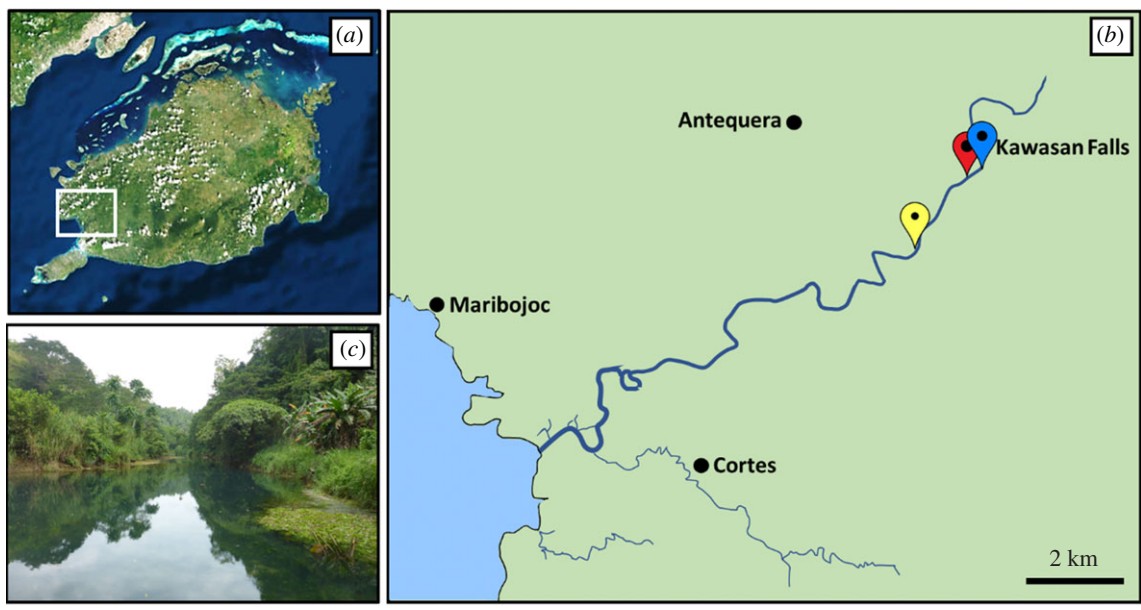

**Figure 1.** Specimen collection site. (*a*) Bohol Island, Philippines. (*b*) Boxed region from (*a*) showing an overview of the Abatan River system. (*c*) Collection site location. Yellow pin, Lozouet & Plaziat [6] station M39 (9°45.2′ N, 123°56.0′ E); red pin, collection site in this study (9°45′56.1″ N 123°56′39.3″ E); blue pin, Kawasan Falls. (Online version in colour.)

**Table 1.** Specimen details.

| specimen # | holding institution | accession # | status | reference |
|---|---|---|---|---|
| PMS-4312Y[a] | Academy of Natural Sciences, Philadelphia, USA | ANSP A477140 | Holotype | figure 3 |
| PMS-4314K[a] | Academy of Natural Sciences, Philadelphia, USA | ANSP A477141 | Paratype, MicroCT | figures 3 and 4 |
| PMS-4134W[a] | Academy of Natural Sciences, Philadelphia, USA | ANSP A477141 | Paratype | figure 3 |
| PMS-4316M | Ocean Genome Legacy Center, Massachusetts, USA | A34420 | DNA Voucher | — |
| PMS-4316M | Ocean Genome Legacy Center, Massachusetts, USA | A34423 | DNA Voucher | figure 5 |

[a]The holotype will be transferred to the National Museum of the Philippines pending completion of an ongoing reorganization.

tube that is secreted onto the burrow walls. This tube surrounds the animal from the posterior edge of the valves to the burrow aperture. The burrow aperture can be sealed by a pair of calcareous, paddle- or feather-like structures, known as the pallets, which are unique to Teredinidae [2,4].

All teredinids described to date are obligate wood-borers that can complete all or part of their entire life cycle within wood [2,4,5]. This includes the giant, chemoautotrophic shipworm, *Kuphus polythalamius* Linnaeus 1767, which has recently been shown to settle on and reach maturity in wood prior to entering the sediment [5]. Remarkably, Lozouet & Plaziat [6] reported a shipworm species, provisionally identified as *Spathoteredo* sp., but known by the locals as 'antingaw', living in a 'mudstone' cliff along the Abatan River, Bohol, Philippines. Here we describe the divergent morphology, habits and phylogenetic origin of this new shipworm and on this basis propose the new genus and species *Lithoredo abatanica*.

## 2. Material and methods

### (a) Specimen collection
Specimens of *L. abatanica* and *Nausitora* sp. were collected from deposits of soft limestone and driftwood, respectively, in the Abatan River (Antequera, Bohol, Philippines), at a depth of less than 2 m, as part of a Philippine Mollusk Symbiont (PMS)

International Cooperative Biodiversity Group (ICBG) expedition. The specimen location map (figure 1) was produced and approximate coordinates were determined using the National Oceanographic and Atmospheric Administration Map Viewer (https://www.nauticalcharts.noaa.gov/ENCOnline/enconline. html). Salinity was measured at the specimen collection site using a refractometer at both low and high tide. Teredinid specimens were carefully extracted from their burrows using a hammer and chisel. All individuals were photographed and specimens for microscopy and taxonomy were fixed in 4% formaldehyde solution freshly prepared from paraformaldehyde, washed and dehydrated through a graded ethanol series (30%, 50% and 70%, 30 min per wash, ×2 washes) with final storage in 70% ethanol. Additional specimens for molecular phylogenetic analysis were fixed and stored in 70% ethanol. Morphological features of soft tissues and calcareous structures were imaged using the Keyence VHX-6000 digital microscope (Osaka, Japan). All specimens were initially assigned unique specimen and lot numbers ('PMS' code) as part of the PMS-ICBG field collection. Type material deposited with the Academy of Natural Sciences of Philadelphia and Ocean Genome Legacy were provided with additional institutional accession numbers. Accession details are provided in table 1.

### (b) Scanning electron microscopy
Scanning electron microscopy (SEM) was used to determine fine-scale shell valves features for the taxonomic description and the

presence of bacteria associated with the gill. For SEM imaging of soft tissues and hard parts, samples were dissected, critical point dried using the Samdri PVT-3D critical point dryer (Tousumis, Maryland, USA), mounted on a standard aluminium SEM stub and coated with platinum to a thickness of 5 nm using the Cressington 208 HR High Resolution Sputter Coater (Cressington Scientific Instruments, Hertfordshire, UK). Samples were imaged on the Hitachi S-4800 field emission scanning electron microscope. Gill tissues were manually fractured using a razor blade prior to coating to expose bacterial symbionts.

## (c) Micro-computed tomography

Micro-computed tomography was used to visualize internal anatomy and three-dimensional spatial arrangement of soft tissues for the taxonomic description. Specimen PMS-4314 K was stained for 20 days in 10% iodine prior to imaging. Imaging was performed using a SkyScan 1173 MicroCT scanner (Bruker MicroCT, Kontich, Belgium) equipped with a Hamamatsu 130/300 tungsten X-ray source and a FlatPanel Sensor camera detector with $2240 \times 2240$ pixels. Scanning parameters were as follows: source voltage = 60 kV; source current = 100 µA; exposure time = 900 ms; frames averaged = 3; frames acquired over $180° = 960$; filter = no; binning = no; flat field correction = activated; scanning time = (x3) 00:28:14; number of connected scans = 3; and, random movement = ON (10). Reconstruction of the raw data was accomplished using the software provided with the scanner (NRECON 1.6.6.0, Bruker MicroCT, Kontich, Belgium). The following settings were employed to enhance image contrast and to compensate for ring and streak artefacts: smoothing = no, ring artefact correction = 4 and beam hardening correction = activated. Reconstructed scans were then analysed using CTVox (Bruker MicroCT, Kontich, Belgium).

## (d) Energy dispersive X-ray elemental analysis and X-ray diffraction mineralogical analysis

Elemental analysis was performed to determine the mineralogical composition of both bored and ingested rock. Transverse sections were cut through the visceral mass of three animals (lot PMS-4174P) using a fine razor blade and analysed using the Hitachi S-4800 field emission scanning electron microscope. Sections were scanned at a working distance of 14.8 mm and an accelerating voltage of 15 kV on the secondary electron detection using the EDX Element EDS Analysis System and accompanying Element EDS Analysis Software Suite. For X-ray diffraction analyses, intestinal tissue from three animals (lot PMS-4162M) and rock samples (PMS-4315 L) were ground to an even grain size in an agate pestle and mortar in 90% ethanol, spread across a microscope slide and allowed to air-dry for 24 h. Samples were analysed on the Rigaku Miniflex 600 X-ray Diffractometer (Tokyo, Japan) with a Cu X-ray tube using the following parameters; 40 kV, 15 mA, start degree 22°, stop degree 50°, interval step 0.005°, speed $0.5° \, s^{-1}$. Mineralogical profiles were then identified using PDXL integrated X-ray powder diffraction software version 2.8 (Tokyo, Japan).

## (e) DNA extraction and amplification

DNA was extracted from the siphonal tissue of two *L. abatanica* (lot PMS-4316M) specimens, using the DNeasy Blood & Tissue kit (Qiagen, Hilden, Germany) as in [7]. Approximate purity, concentration and yield of DNA were determined by UV spectrophotometry. Genomic DNA was cryopreserved at −80°C and archived at the Ocean Genome Legacy Center of New England Biolabs, Northeastern University, Nahant, MA, USA (accession number in table 1). The small (18S) and large (28S) subunit nuclear rRNA genes were amplified from the resultant DNA

preparation by polymerase chain reaction (PCR). Amplification reactions were prepared using 12.5 µl of polymerase solution (One*Taq*, New England Biolabs, Ipswich, Massachusetts), 0.5 µl of each primer (10 mM), 1–2 µl DNA template (10–20 ng µl⁻¹), brought to a total volume of 25 µl with purified water. Fragments of the small (18S) and large (28S) subunit nuclear rRNA genes were amplified using the primer pairs 18S-EukF (5′-WAY-CTG-GTT-GAT-CCT-GCC-AGT-3′) and 18S-EukR (5′-TGA-TCC-TTC-YGC-AGG-TTC-ACC-TAC-3′) [8]; 28S-NLF184-21 (5′-ACC-CGC-TGA-AYT-TAA-GCA-TAT-3′) and 28S-1600R (5′-AGC-GCC-ATC-CAT-TTT-CAG-G-3′) [9], resulting in amplicons of approximately 1686 and 1416 base pairs, respectively. The PCR amplification proceeded as follows: an initial denaturation step of 94°C for three minutes, followed by 35 cycles with a denaturation step of 94°C for 20 s, an annealling step of 64°C for 40 s for the 18S and 63°C for 30 s for 28S, an extension step of 68°C for 60 s and a final extension of 68°C for 5 min. All reactions were performed on a PTC-200 Thermal Cycler (MJ Research, Quebec, Canada).

For each template, three separate amplicons, all produced under identical conditions, were pooled, cleaned and concentrated using the Zymo Clean & Concentrator Kit (Zymo, Research, Irvine, CA). Resulting products were sequenced using external and internal primers [9] bidirectionally on a 3730xl DNA Analyzer (Life Technologies, Grand Island, NY) using the Big Dye Terminator 3.1 Cycle Sequencing Kit (Life Technologies, Grand Island, NY) at New England Biolabs. Primary sequence data from the small (18S) and large (28S) subunit nuclear rRNA genes were submitted to GenBank (NCBI) under accession numbers MN028384 and MN027962, respectively.

## (f) Phylogenetic analysis

As the 18S and 28S rRNA gene sequences from both specimens (lot PMS-4316M) were identical, sequence data from only one specimen was concatenated and aligned as in Distel *et al.* [9] and Shipway *et al.* [7] with sequences representing 11 of 17 recognized genera of Teredinidae [2,5,7,10,11] including: *Bankia* Gray 1842, *Dicyathifer* Iredale 1932, *Kuphus* Guettard 1770, *Lyrodus* Gould 1870, *Nausitora* Wright 1864, *Neoteredo* Bartsch 1920, *Spathoteredo* Moll 1928, *Teredo* Linnaeus 1758, *Teredora* Bartsch 1921, *Teredothyra* Bartsch 1921, and *Tamilokus* Shipway, Distel & Rosenberg, 2018, as well as 39 bivalve reference taxa. Phylogenetic analysis was performed using MRBAYES v. 3.2.6 [12] with the $GTR + I + \Gamma$ nucleotide substitution model as selected using JMODELTEST v. 2.1.10 [13] with Akaike Information criterion. *Placopecten magellanicus* was specified as the outgroup, the chain length was set to 5 million, subsampling every 2000 generations and discarding the first 20% of the results as burn-in.

## 3. Results

Multiple specimens of *L. abatanica* were collected from carbonate limestone exposed in the bed and low banks of the Abatan River, Antequera, Bohol, Philippines (figures 1 and 2). This location is approximately 2 km upstream from station M39 of Lozouet & Plaziat [6], the only other confirmed location for this species (figure 1). Water samples taken at the site measured less than 0.5 ppt salinity at both low and high tides. Specimens ranged from 5.5 to 105.4 mm in total body length (measurements taken from living animals prior to fixation; figure 3*a*–*c*). Numerous nestling macroinvertebrate taxa were recorded occupying vacated burrow galleries of *L. abatanica*, including crabs, shrimp, limpets, snails (Neritidae), mussels (Mytilidae) and polychaetes (electronic supplementary material, figure S1). Although teredinids

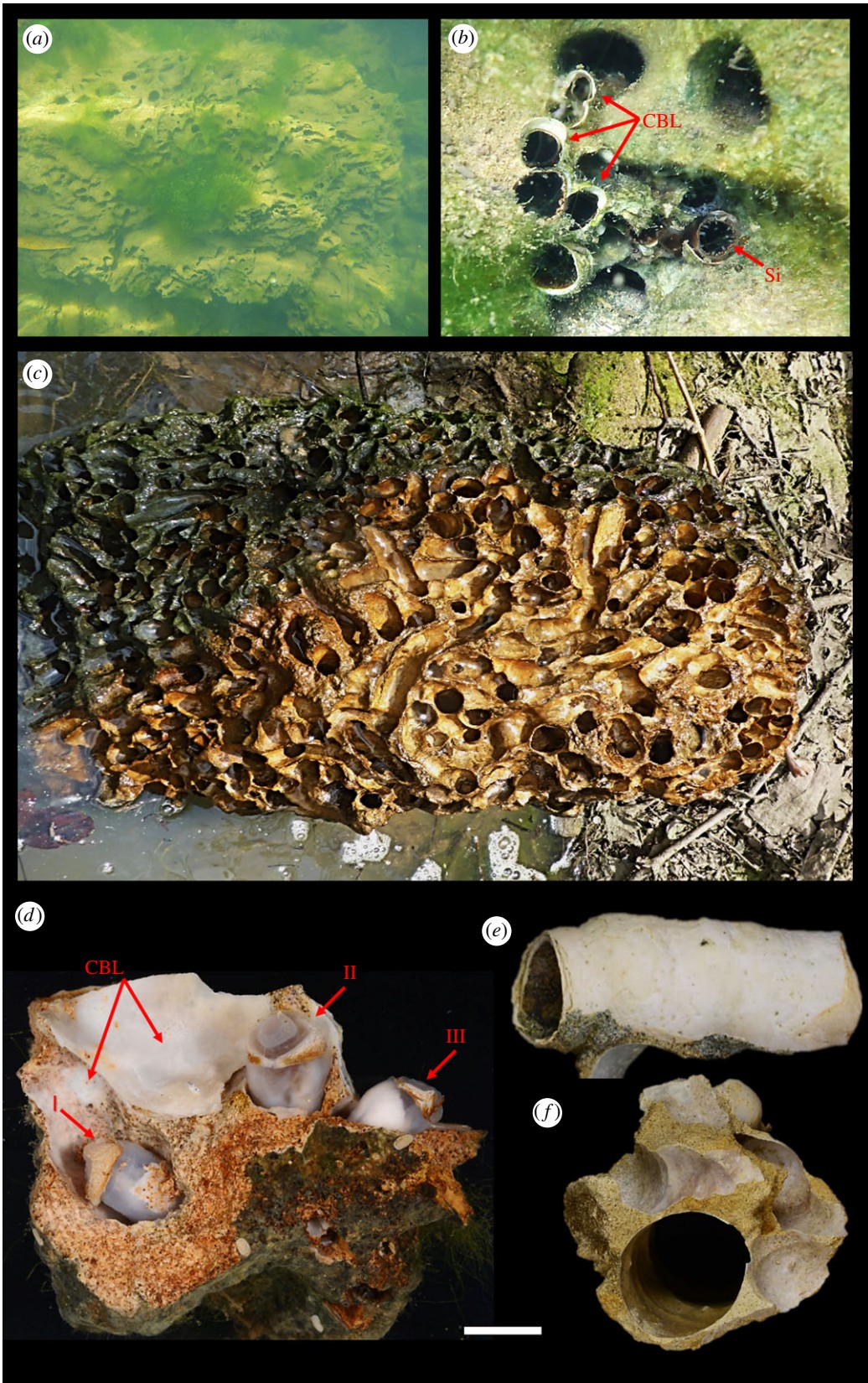

**Figure 2.** Freshwater macrobioerosion resulting from the burrowing activity of *Lithoredo abatanica*: (*a*) carbonate limestone bedrock showing bioerosion from *L. abatanica*; (*b*) siphons of living shipworms extending from calcareous burrow linings within bedrock; (*c*) section of carbonate limestone featuring galleries formed by the closely spaced empty burrows of *L. abatanica*; (*d*) a small fragment of limestone bedrock (specimen PMS-4174P) showing the anterior ends of three living specimens of *L. abatanica* (labelled I–III); (*e*) calcareous tube produced by *L. abatanica* removed from burrow; (*f*) cross section through calcareous tube within limestone. Scale bar = 10 mm. CBL, calcareous burrow lining; Si, siphon. (Online version in colour.)

identified as members of the genus *Nausitora* were found inhabiting submerged wood in the surrounding area, no teredinids matching the characteristics of those found in carbonate limestone were present in any wood examined.

### (a) MicroCT and scanning electron microscopy

Scanning electron micrograph images of the shell valves of *L. abatanica* reveal the absence of the sharp fine-scale denticulation common to wood-boring Teredinidae. Instead the shell

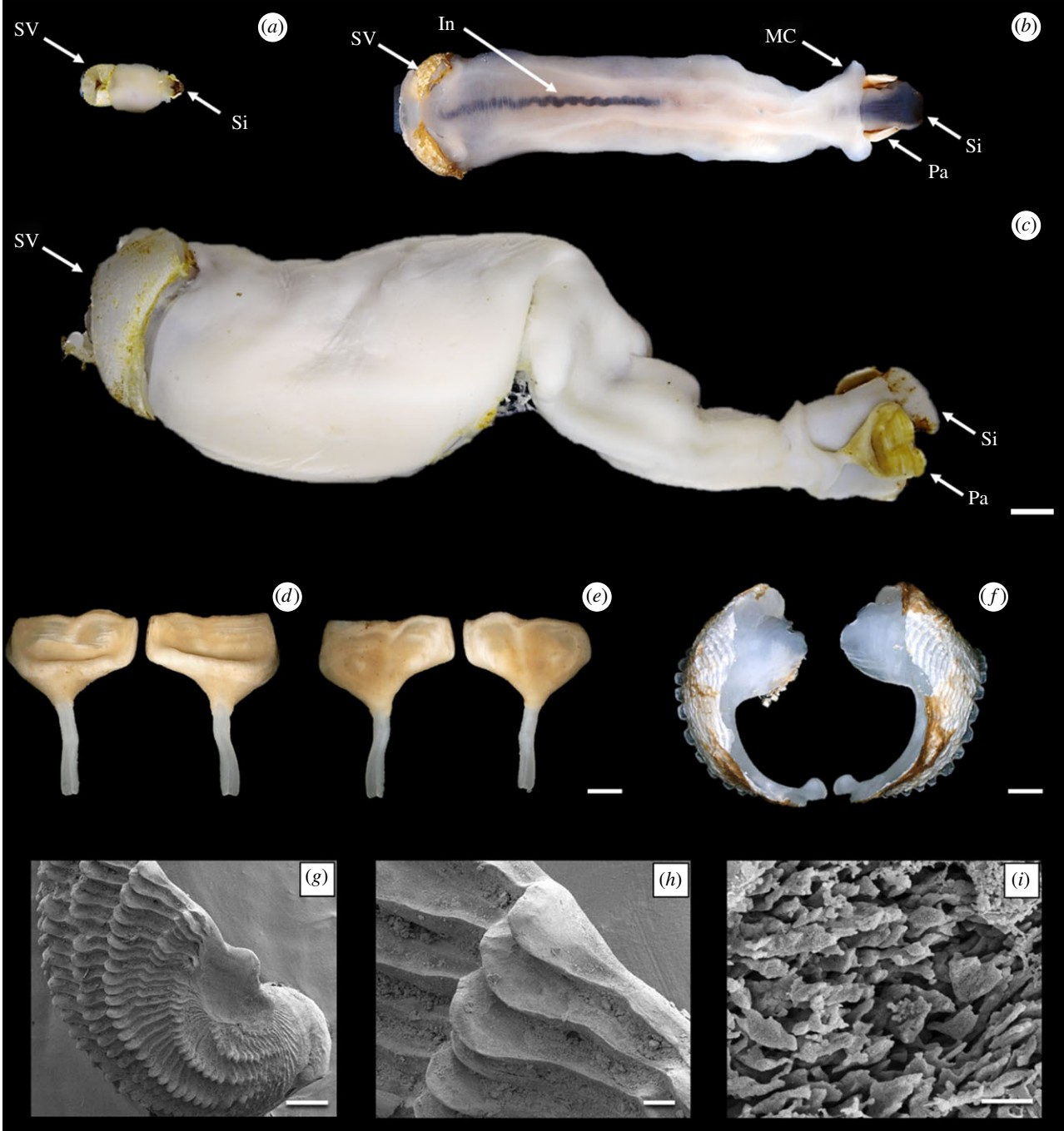

**Figure 3.** Morphology of *Lithoredo abatanica*: (*a*) juvenile specimen (PMS-4313H); (*b*) small adult specimen (PMS-4134 W); (*c*) large adult specimen (holotype PMS-4312Y); (*d*) pallet pair outer face; (*e*) pallet pair inner face; (*f*) shell valves; (*g*) scanning electron micrograph of shell valve; (*h*) magnified region from (*g*) showing valve denticulation; (*i*) magnified region from (*h*). In, intestine; MC, mantle collar; Pa, pallet; Si, siphon; SV, shell valve. Scale bar (*a*−*c*) = 5 mm, (*d*−*f*) = 1 mm, (*g*−*i*) = 200 μm, 100 μm and 5 μm respectively. (Online version in colour.)

denticles of *L. abatanica* are large, broad, blunt and spatulate (figure 3*g*−*i*). A MicroCT virtual model of *L. abatanica* (figure 4), revealed several unusual features including the absence of a caecum (figure 4*a*−*e*), gills that extend the entire length of the animal (electronic supplementary material, figure S6*a*) and an elongated intestine that extends deep into the anal canal (figure 4*a*−*f*). Two-dimensional representations of the MicroCT 3D rendered model of *L. abatanica*, including major anatomical structures and transverse sections are shown in figure 4 and a three-dimensional virtual video of the anatomy and morphology *L. abatanica* is shown in electronic supplementary material, figure S3. Scanning electron micrographs of fractured gill tissues reveal the presence of abundant bacterial cells (electronic supplementary material, figure S5).

## (b) Gut content analysis

Energy dispersive X-ray analysis (EDX) performed on transverse sections through the visceral mass of three specimens of *L. abatanica* show that the intestinal content is similar in elemental composition to that of the rock substrate in which the animals bore, containing the same elements (C, O, Si, Al, Ca, Mg and Fe) in similar proportions. Contrastingly, tissues adjacent to the intestine, including the epithelial intestinal membrane, gonad and mantle, displayed peaks only for C, N, O and P and the negative control (sample mounts only, with no added sample) displayed peaks for C and O only (electronic supplementary material, figure S6*a*−*c*). X-ray diffraction (XRD) analysis of intestinal content from three animals also matched the mineralogical

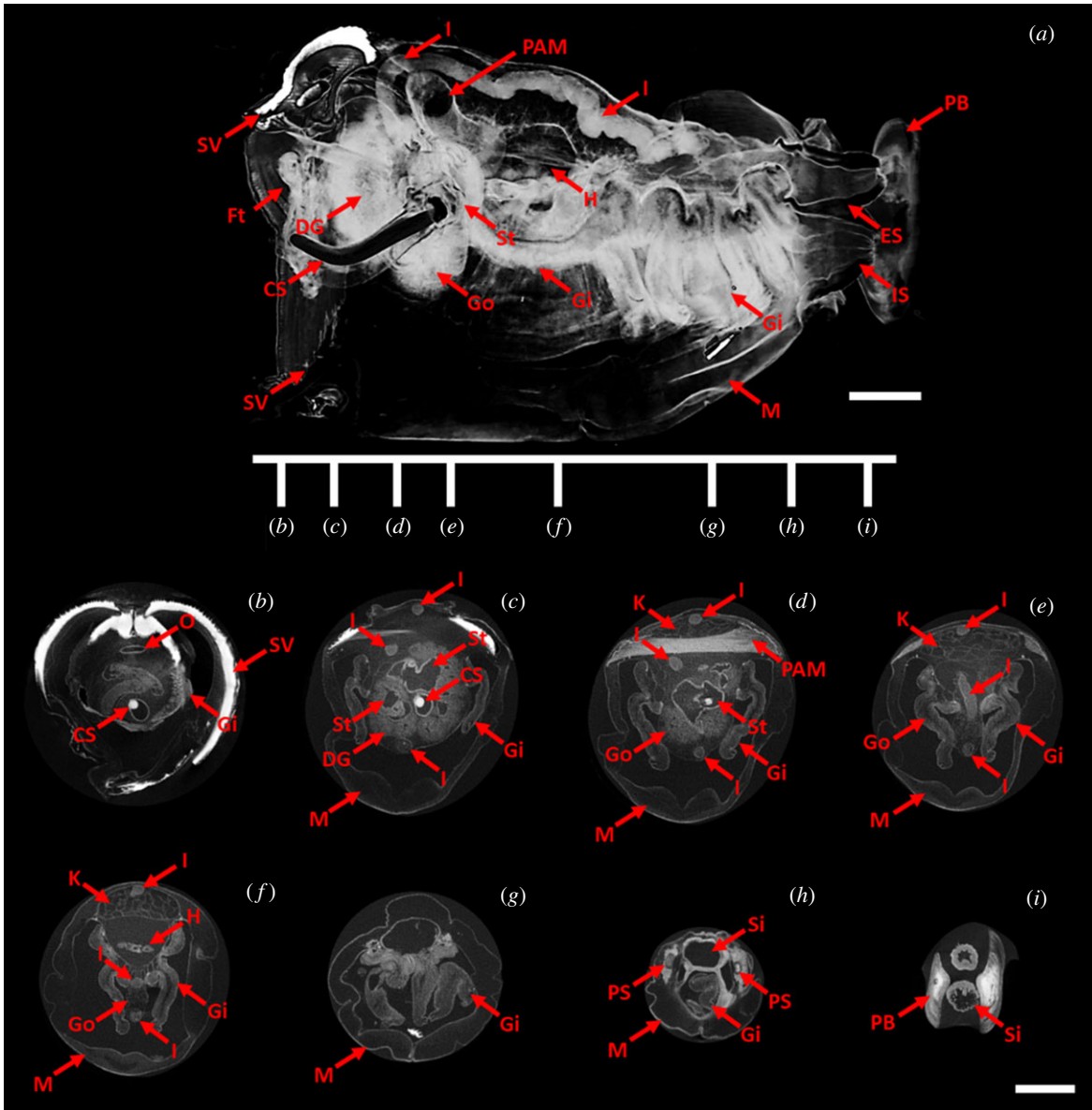

**Figure 4.** Anatomy of *Lithoredo abatanica* (specimen PMS-4314 K): two-dimensional representations (virtual sections) from a three-dimensional render of *L. abatanica* generated by micro-computed tomography. (*a*) Longitudinal virtual section in the plane defined by the anterior–posterior (left–right) axis and the dorsal–ventral (top–bottom) axis. (*b*–*i*) Transverse sections corresponding to locations along the anterior posterior axis indicated by the linear map in (*a*). CS, crystalline style; DG, digestive gland; ES, excurrent siphon; Ft, foot; Gi, gill; Go, gonad; H, heart; I, intestine; IS, incurrent siphon; K, kidney; M, mantle; O, oesophagus; PAM, posterior adductor muscle; PB, pallet blade; PS, pallet stalk; Si, siphon; St, stomach; SV, shell valve. Scale bar = 2.5 mm. (Online version in colour.)

profiles of the rock samples into which the animals were boring. Both intestinal content and rock samples were shown to match a mineralogical reference standard for calcite (C Ca0.936 Mg0.064 O3, 9001297) using PDXL software (electronic supplementary material, figure S6*d*).

## (c) Phylogeny

Bayesian analysis of concatenated small (18S) and large (28S) nuclear rRNA gene sequences places *L. abatanica* within the Teredinidae, in a well-supported clade containing members of the genera *Dicyathifer*, *Tamilokus* and *Teredora*, but separated from these taxa by a branch length comparable to those separating most accepted teredinid genera. The phylogenetic position of *L. abatanica* among the Teredinidae is shown in figure 5 and electronic supplementary material, figure S4.

## (d) Systematics

**Family Teredinidae Rafinesque, 1815**
***Lithoredo* Shipway, Distel & Rosenberg, gen. nov.**
urn:lsid:zoobank.org:act:CEA21AF7-40AC-44A4-B92B-D42EC018C22A

Type species: *L. abatanica* sp. nov.

**Diagnosis:** Pallets pentagonal with deep 'thumbnail'-like depression on outer surface and slight medial ridge on inner surface (figure 3*d,e*); valves lack fine-scale denticulation, but feature large, undulating ridges of spatulate denticles across the outer face (figure 3*f*–*i*); valves are anteroposteriorly narrow, and the height is approximately two times longer than width; additionally, valves are almost entirely composed of the anterior slope, with a slight posterior slope in adults and slightly more pronounced posterior slope in juveniles; caecum indistinct or absent (figure 4*a*); mantle thick and white to

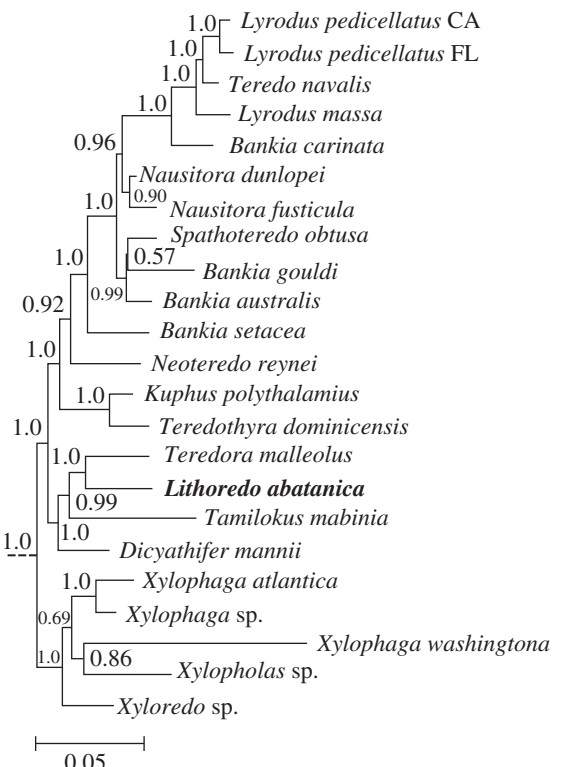

**Figure 5.** Phylogenetic position of *Lithoredo abatanica* within Teredinidae. A sub tree excerpted from a Bayesian analysis of the concatenated 18S and 28S nuclear rRNA gene sequences obtained from specimen PMS-4316M. The full tree is presented in electronic supplementary material, figure S4. Numbers at nodes indicate posterior probabilities. Scale bars denote nucleotide substitutions per site.

cream coloured (figure 3*a–c*); three rows of papillae on both excurrent and incurrent siphon apertures.

**Etymology:** *Lithoredo*, feminine, a combination of *litho-* (*rock*) and the last two syllables of the Latin noun *teredo* (shipworm).

**Remarks:** *Lithoredo* is most similar to *Teredora* and *Uperotus* in that it possesses a gill that extends the entire length of the animal, from the base of the siphons to the mouth (figure 4*a*). However, *Lithoredo* is easily distinguished from these genera based on pallet and valve morphology (described below) and the absence of a caecum. A comparison of characters between *Lithoredo*, *Tamilokus*, *Teredora* and *Uperotus*—phylogenetically the most closely related genera—is provided in electronic supplementary material, table S1.

 *Lithoredo* gen. nov may be distinguished from other genera within the family Teredinidae based on the calcareous structures. The valves of *Lithoredo* are unique among the family, with broad, undulating ridges of spatulate denticles (figure 3*f–i*) replacing the small acute denticles found in all currently described genera except *Kuphus*, where denticles are large, blunt and featureless. The simple, solid, paddle-shaped pallets of *Lithoredo* may be distinguished from the pallets of *Bankia*, *Nausitora*, *Nototeredo* Bartsch 1923, and *Spathoteredo* by the lack of segmentation; from *Lyrodus*, *Teredo*, *Zachsia* Bulatoff & Rjabtschikoff 1933, and *Nivanteredo* Velásquez & Shipway 2018 by the absence of a periostracum; from *Teredothyra* by the absence of a divided cup; from *Bactronophorus* Tapparone Canefri 1877 by the absence of a dagger-like extension; from *Dicyathifer* and *Kuphus* by the absence of a distinct medial ridge on the outer face; from *Neoteredo* by the narrow blade

and slight depression; from *Uperotus* Guettard 1770 by the smooth surface of the thumbnail-like depression on the outer of the pallet, lacking concentric or radiating ridges; and, from both *Psiloteredo* Bartsch 1922, and *Teredora*, the pallets of which feature rounded, paddle-shaped blades compared with the angular, pentagonal profile of the blade in *Lithoredo*.

### *Lithoredo abatanica* Shipway, Distel & Rosenberg, sp. nov.
urn:lsid:zoobank.org:act:16631201-18E9-45D4-808A-67B7AE7FFB3D

**Synonymy:** *Spathoteredo* sp. Lozouet & Plaziat [6]: 91, 93; pl. 9, fig. 14–16; and pl. 10, fig. 1–3.

**Type material:** Holotype PMS-4312Y, measuring 105.4 mm in total body length, collected on the 20th of August 2018 by J. Reuben Shipway, Marvin A. Altamia and Melfeb Chicote, fixed in 4% formaldehyde and stored in 70% ethanol; the holotype is currently held at the Academy of Natural Sciences of Philadelphia (ANSP) and will be deposited in the National Museum of the Philippines in Manila pending completion of an ongoing reorganization. Paratypes are deposited at ANSP and the Ocean Genome Legacy Center of New England Biolabs, Northeastern University (OGL). For catalogue numbers assigned by holding institutions see table 1.

**Type locality:** Abatan River, Antequera, Bohol, Philippines.

**Comparative Material:** The following material was examined for comparison: *Teredora malleolus* (Turton, 1822) MCZ 350541 (three specimens); *T. princesae* (Sivickis, 1928) MCZ 232090 (five specimens); *Uperotus clava* (Gmelin, 1791) MCZ-238009 (three specimens); *U. panamensis* (Bartsch, 1922) MCZ 357824 (four specimens); and *Tamilokus mabinia* Shipway, Distel & Rosenberg, 2019 ANSP A476699 (four specimens).

**Habitat:** Oligohaline-freshwater, infaunal stone-borer.

**Description:** Pallets pentagonal and white to cream coloured, formed of a broad thick blade featuring a flat distal margin, a deep 'thumbnail'-like depression and a simple, translucent stalk (figure 3*d,e*; electronic supplementary material, figure S2); valves are thick, and lack fine-scale denticulation; instead, valves feature large, undulating ridges across the outer face formed of rows of large, blunt, spatulate denticles (figure 3*f–i*, electronic supplementary material, figure S2); inner surface of valves feature thickened apophysis and dorsal/ventral condyles; labial palps pronounced and free; large, globular-shaped stomach (figure 4*a,c,d*); caecum indistinct or absent (figure 4*a,c–e*); intestine encircles style sac, digestive glands and gonad (figure 4*e*), makes a simple loop over the posterior adductor muscle and then extends posteriorly into the midpoint of the anal canal (figure 4*a,c–f*); anal canal is open and does not retain faeces (figure 4*a*); heart located medially (figure 4*a,f*); gills are broad and blade-like, with well-separated right and left demibranchs, extending the entire length of the animal, from the base of the siphons to the mouth (figure 4*a,d–g*); both incurrent and excurrent siphons feature three rows of papillae; inner row of papillae on incurrent siphon are larger and longer than proceeding two rows; aperture of calcareous tube large, forming distinct figure-of-eight surrounding both the excurrent and incurrent siphons (figure 2*b*); thick mantle, white to cream in colour (figure 3*a*).

**Etymology:** *abatanica* (adjective) refers to the type locality, the Abatan River, Bohol, Philippines.

*Proc. R. Soc. B* **286**: 20190434

**Remarks:** *L. abatanica* is superficially similar to the illustration of *Teredo ancilla* in Barnard (1964) [14], but differs in several important ways: (1) the siphons of *T. ancilla* are separate along their entire length and feature few inconspicuous papillae, conversely the siphons of *L. abatanica* are almost entirely united (up to three-fourth of their entire length) and feature multiple rows of large papillae on both the incurrent and excurrent aperture; (2) the intestine is visible through the thin mantle on the ventral surface in *T. ancilla*, whereas the ventral mantle in *L. abatanica* is thick and opaque, and the intestine is only visible dorsally; and (3) *T. ancilla* was found in mangrove wood, not rock.

# 4. Discussion

Herein, we present the description of a new genus and species of teredinid from an oligohaline-freshwater river in the Philippines. Remarkable for burrowing into carbonate limestone as opposed to woody substrates (figure 2a−d), *L. abatanica* is anatomically and morphologically distinct from all extant taxa and raises the total number of genera in this ecologically, economically and historically important family to 18.

Several taxonomic characters identify *L. abatanica* as a new genus and species, including calcareous pallets distinct from those of described genera (figure 3d,e), unique valves featuring undulating ridges across the boring surface (figure 3f−i), the absence of a caecum and the extension of the intestine into the mid-point of the anal canal (figure 4a). In addition to these unique anatomical and morphological features, phylogenetic analysis based on concatenated small (18S) and large (28S) nuclear rRNA gene sequences reveal well-supported relationships between *Lithoredo* and several basal taxa within the Teredinidae (figure 5). *Lithoredo* is sister to *Teredora*, one of only two described genera (*Uperotus* being the other) also featuring gills that extend the entire length of the animal, from the base of the siphons to the mouth [2].

All previously described shipworm genera are obligate wood-borers [2,3], including the giant chemoautotrophic shipworm *Kuphus polythalamius*, which has been shown to inhabit wood early in its life cycle and is now thought to transition to sediments after initial settlement and metamorphosis on wood [5]. However, it seems unlikely that *L. abatanica* makes a similar transition from wood to rock. First, careful examination of submerged wood found in close proximity to bored carbonate limestone contained no evidence of *L. abatanica*, although this wood contained numerous specimens of a wood-boring *Nausitora*. Second, specimens of *L. abatanica* less than 5 mm in length were found burrowing in stone, suggesting that they were recently metamorphosed. Thirdly, anatomical and morphological adaptations that facilitate wood-boring and wood-digestion, including fine shell dentition and a wood-storing caecum, are absent in both large and small specimens of *L. abatanica*. Conversely, *L. abatanica* is well adapted for rock-boring, as evidenced by the thick valves with deep, excavating ridges and the large posterior adductor muscle which attaches to the posterior slope of the valves (electronic supplementary material, figure S2). Finally, it is hard to imagine a physical mechanism by which *L. abatanica* might transition from wood to rock, similar to *K. polythalamius* transitioning from wood to sediments [5]. Soft sediments can conform to the surfaces of submerged wood, allowing *K. polythalamius* to transition from wood to

sediments without disturbing the continuity of the burrow and burrow lining. However, this is not possible for the hard rock surfaces in which *L. abatanica* burrows. Thus, any transition between wood and stone would be restricted to small infrequent contact patches between wood and stone. This mechanism is not consistent with the dense pattern of burrows observed over the rock surfaces and sparsity of wood observed at the collection site.

Analysis of the gut content of *L. abatanica* demonstrates that the ingested material has a mineral content closely matching the composition of the rock in which the animals burrow (electronic supplementary material, figure S6) indicating that this species ingests rock as it burrows. Although many other invertebrate species are known to burrow in stone [15], we are not aware of other species that burrow in stone by ingesting the substrate. We suggest that this unusual habit is a consequence of this species having evolved from a wood-feeding ancestor, since the mechanism by which shipworms burrow in wood involves both ingestion and digestion of the excavated wood [2,16]. However, it remains an open question whether *L. abatanica* derives any degree of sustenance from the limestone or whether the ingested rock may contribute to nutrition in other ways. For example, the ingested rock might contribute to trituration of ingested materials as do the gizzard stones of lithophagous birds and reptiles [17,18]. The observed dark colour of the gut content (e.g. figure 3b) of *L. abatanica* contrasts with the light-coloured rock and suggests that in addition to mineral matter, these animals may also ingest pigmented particulate material from the water column, such as planktonic algae, bacteria or terrestrial plant matter. Indeed, the large size of the gills and labial palps are consistent with a capacity for planktotrophy [2,4]. Alternatively, *L. abatanica* may feed on deposited materials such as green algae (e.g. figure 2a,b), cyanobacteria and other microorganisms that typically amass on and within freshwater carbonate limestone [19]. Finally, like other members of the family Teredinidae, *L. abatanica* may utilize a symbiont-dependent mode of nutrition. Indeed, scanning electron micrographs reveal that the gills of *L. abatanica* are densely populated by bacterial symbionts (electronic supplementary material, figure S5), however, it remains to be determined whether these bacteria contribute to the nutrition of *L. abatanica* via xylotrophic or chemoautotrophic endosymbioses as observed in other shipworms [20,21]. These questions are outside the scope of the current work but are the subject of an ongoing investigation.

The burrowing habits of *L. abatanica* may also have important implications with respect to bioerosion. Previously, macrobioerosion has been considered a process exclusively limited to coastal marine to estuarine environments and has only recently been documented in freshwater systems, such as the rock-boring mussel *Lignopholas fluminalis* (Blanford, 1867) [21−23] and chironomid and trichopteran insect larvae [22]. Although well studied in marine and estuarine realms, freshwater settings have received limited attention and the potential impacts of freshwater organisms on the erosion of cohesive substrates has been underappreciated [23]. Our findings of a substantial population of shipworms burrowing into the limestone bedrock of the Abatan River, Philippines, greatly expand knowledge of a key ecological process in a previously poorly studied environment.

The burrowing habit of *L. abatanica* may also play a role in shaping its ecosystem and creating new habitats. Shipworms

are notorious for their ability to overwhelmingly colonize and rapidly degrade lignocellulosic materials [24]. In mangrove ecosystems, these animals account for approximately 70% of wood turnover [25], and the labyrinth of excavated tunnels provide refuge for fish and numerous marine invertebrates [26,27]. In this regard, the ecological impact of *L. abatanica* is consistent with that of other teredinids, as evidenced by the significant colonization of bedrock (figure 2*a*–*d*), and the high fragmentation of this calcitic material strewn across the bank of the Abatan River. Additionally, several invertebrate taxa were found residing within the intricate network of derelict burrows (electronic supplementary material, figure S1). In this system, we suggest *L. abatanica* is the dominant ecosystem engineer; the substantial bioeroding behaviour drives niche creation, increases habitat complexity for a range of species assemblages and likely alters the course of the Abatan River.

Teredinid bivalves are largely responsible for elongate borings found in fossil marine and marginal marine woodgrounds (i.e. the ichnospecies *Teredolites longissimus*). Notably, where mechanical and biodegradational processes have destroyed evidence of the original wood, *Teredolites*-like casts and/or linings often provide the only evidence for the pre-existence of marine woodgrounds [28]. Our finding highlights that such teredinid-generated *Teredolites*-like ichnofossils are not exclusive indicators of marine palaeo-woodground ecosystems but may also reflect freshwater riverine ecosystems and non-xylic habitats (figure 2*d*–*f*). Additionally, rocks featuring extensive boreholes were considered one of the primary diagnostic features of shallow marine palaeoenvironments. *Lithoredo abatanica*, along with the recent discovery of the rock-boring mussel *Lignopholas* in Kaladan River, Myanmar [22], provides further evidence that rocks featuring macroborings are found in freshwater habitats and cannot be considered absolute indicators of marine palaeo-ecosystems (figure 2*a*–*d*,*f*).

Finally, we note that *L. abatanica* may be a species of conservation concern deserving of special consideration for habitat protection. Currently, the known distribution range is limited to a very restricted area along the Abatan River, Bohol, Philippines, bounded by the site described here and station M39 of Lozouet & Plaziat [6], located approximately 2 km downstream (figure 1). The observed salinities range from marginally brackish (salinity 1–2 ppt) at station M39 to freshwater less than 0.5 ppt at the location reported here, suggesting a narrow salinity tolerance of oligohaline-freshwater for this species. Furthermore, the unusual life habits of *L. abatanica* suggest a limited capacity for distribution to new locations. As the trophic strategy of this species remains obscure and possibly unique, and no anatomically, morphologically or ecologically similar species are known, this species should be a focus of special research and conservation efforts.

## 5. Conclusion

Herein, we describe *L. abatanica*, a new genus and species of shipworm, found in carbonate limestone bedrock in an oligohaline-freshwater river in the Philippines. We demonstrate that *L. abatanica*, in contrast to previously described shipworm species, is probably not an obligate wood-borer and lacks the anatomical and morphological specializations typically associated with wood-boring and wood-digestion in other taxa. Further, we show that *L. abatanica* burrows into and ingests limestone, which accumulates in the guts of animals and is expelled from the siphons as fine-grained particles. This strategy of burrowing into rock by ingestion is, to our knowledge, unique among the animal kingdom.

These findings extend and reinforce the recent conclusion that rock-boring macrobioerosion is not restricted to marine environments, and highlights the significance of *L. abatanica* as an ecosystem engineer, driving niche creation and increasing habitat complexity for a host of oligohaline-freshwater organisms. In addition, we suggest these findings are of significant importance to palaeontology, and demonstrate that fossils attributed to teredinids are not exclusive indicators of marine palaeo-woodground ecosystems and may also reflect freshwater riverine ecosystems and non-xylic habitats.

Ethics. All Philippine specimens used in this study were obtained using Gratuitous Permit 0140-17 issued by DA-BFAR.

Data accessibility. The datasets supporting this article have been uploaded as part of the electronic supplementary material S3.

Authors' contributions. J.R.S. collected specimens, carried out laboratory work, participated in data analysis, participated in the design of the study and drafted the manuscript; M.A.A. collected specimens and participated in data analysis and drafted the manuscript; M.G.H., G.P.C. and G.R. contributed to the study design; and D.L.D. conceived, designed and led the study and drafted the manuscript. All authors gave final approval for publication and agree to be held accountable for the work performed therein.

Competing interests. We declare we have no competing interests.

Funding. The research reported in this publication was supported by Fogarty International Center of the National Institutes of Health Award U19TW008163 (to M.G.H., G.R., G.P.C. and D.L.D.), by National Science Foundation Award IOS1442759 (to D.L.D.).

Acknowledgments. The authors thank William Fowle and Isaac Westfield (Northeastern University) for assistance with the energy dispersive X-ray and X-ray diffraction analysis, Adam Baldinger and Jennifer Lenihan-Trimble (Museum of Comparative Zoology at Harvard University) for use of the Keyence Microscope and Bruker MicroCT, Melfeb Chicote (Marine Science Institute, University of the Philippines) for assistance collecting specimens and Charles Savrda (Auburn University) for helpful comments. Finally, the authors would like to thank Philippe Bouchet (Muséum National d'Histoire Naturelle, Paris) and Takuma Haga (National Museum of Nature and Science, Tokyo) for providing helpful information. Part of this work was completed under the supervision of the Department of Agriculture Bureau of Fisheries and Aquatic Resources, Philippines (DA-BFAR), in compliance with all required legal instruments and regulatory issuances covering the conduct of the research.

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
