## [Reviewer comments · Proceedings of the Royal Society B: Biological Sciences]

Review History

RSPB-2019-0434.R0 (Original submission)

Review form: Reviewer 1 (Ivan Bolotov)

Recommendation

Accept with minor revision (please list in comments)

Scientific importance: Is the manuscript an original and important contribution to its field?

Excellent

General interest: Is the paper of sufficient general interest?

Excellent

Quality of the paper: Is the overall quality of the paper suitable?

Excellent

Is the length of the paper justified?

Yes

Should the paper be seen by a specialist statistical reviewer?

No

Do you have any concerns about statistical analyses in this paper? If so, please specify them explicitly in your report.

No

It is a condition of publication that authors make their supporting data, code and materials available - either as supplementary material or hosted in an external repository. Please rate, if applicable, the supporting data on the following criteria.

Is it accessible?

Yes

Is it clear?

Yes

Is it adequate?

Yes

Do you have any ethical concerns with this paper?

No

Comments to the Author

Dear colleagues,

Oh my God, Fantastic Beasts and Where to Find Them? In the Philippines, of course. A rock-boring and rock-ingesting freshwater shipworm! My God! It looks like an alien creature, not an animal from our planet. Congratulations with this amazing finding! After our discovery of a freshwater rock-boring mussel in Myanmar I didn't think that I could be surprised by anything but now ... now my brain felt as if it was gonna explode.

This manuscript is very well written and concise. As far as I know, it reports one of the greatest discoveries in freshwater malacology and hydrobiology during the last decades, if not the last hundred years. In my opinion, it is fully appropriate for the journal, and could be published after a minor revision.

I have only a few comments as follows:

219: ...using PDXL software Figure 5D => ...using PDXL software (Fig. 5D)

226-257: In a description of the genus-level taxon, you should indicate the type species (i.e. line 225) and provide a brief diagnosis of the genus (i.e. lines 238-243), etymology (lines 244-245), and remarks (lines 258-281). However, the type material (lines 226-231), type locality (line 232), comparative material (lines 233-237), habitat (line 246), and description (lines 247-257) seem to be inappropriate for a description of the genus-level taxon. These paragraphs must be transferred to the species description. Your abatanica will be a nomen nudum if this description will be published in its current form, because your description does not conform the rules of the ICZN: no reference to the type series is provided; the type locality is not indicated; a description is too short and incomplete.

228-231: "Paratypes are deposited at ANSP, at the Marine Science Institute of the University of the Philippines (MSI) and at the Ocean Genome Legacy Center of New England Biolabs, Northeastern University (OGL)." => This statement does not correspond to Table 1. Please list all available paratypes with their voucher codes in this table.

296: from a fresh-water... => from a freshwater...

283: *Lithoredo abatanica* Shipway & Distel, sp. nov. => *Lithoredo abatanica* Shipway & Distel, gen. & sp. nov.

549: Figure 3: The Morphology... => Figure 3: Morphology...

557: Figure 4: The anatomy... => Figure 4: Anatomy...

596-597: Why you want to transfer the whole type series (lines 596-597) or at least the holotype (lines 226-228) from the ANSP to a Philippine museum? I strongly disagree with this idea. From a historical perspective, there are only few countries and few large museums in the world, in which the long-term storage of type collections can be secured, i.e. the United States (ANSP, MCZ, USNM, and others), UK (BMNH), Germany (SMF, ZMB), France (MNHN), Australia (AMS), and some others. For example, I spent a lot of time transferring the holotypes of my new taxa to the US (e.g. to the North Carolina Museum of Natural Sciences), because I am not sure that they will be safe (in a long-term perspective!) in Russian depositories. Additionally, the types will be much more accessible for researchers in the ANSP than in a Philippine museum. Could you deposit the holotype and 2-3 paratypes in Philadelphia? Several paratypes can also be transferred to the National Museum of the Philippines and other depositories. I guess it would be the best solution. Supp. Fig. 1: clam = Mytilidae (freshwater lineage of *Xenostrobus* sp.); gastropod = Neretidae (*Septaria* cf. *luzonica*?). Nice animals. Just for information.

I look forward to read this great paper published!

With kind greetings from frozen Arctic Russia,
Ivan Bolotov

Review form: Reviewer 2 (Takuma Haga)

Recommendation

Major revision is needed (please make suggestions in comments)

Scientific importance: Is the manuscript an original and important contribution to its field?

Good

General interest: Is the paper of sufficient general interest?

Marginal

Quality of the paper: Is the overall quality of the paper suitable?

Poor

Is the length of the paper justified?

Yes

Should the paper be seen by a specialist statistical reviewer?

No

Do you have any concerns about statistical analyses in this paper? If so, please specify them explicitly in your report.

No

It is a condition of publication that authors make their supporting data, code and materials available - either as supplementary material or hosted in an external repository. Please rate, if applicable, the supporting data on the following criteria.

Is it accessible?

Yes

Is it clear?

No

Is it adequate?

No

Do you have any ethical concerns with this paper?

No

Comments to the Author

I apologize for the delay in reviewing the manuscript.

The manuscript describes a novel finding of an eccentric shipworm from “unusual habitat” and tries to discuss its potential impact. While I appreciate the effort of the work and the significance of this description, I think the authors need to improve the resulting discussion suitable for PRSB with providing more concrete data and scenarios those may attract the readers: implications for rock-ingestion and feeding mode still need more data to clarify their nutrient origin, and to do so, the authors need to show data on possible symbionts in their gills, even if it is concluded to be barren; implication for teredinid evolution needs some plausible scenarios from an obligate wood-boring into obligate rock-boring mode of life, but now the authors only showed phylogenetic position of the shipworm and demonstrated that the shipworm is an obligate rock-borer; and implication for trace fossils needs a comparison with adequate ichnotaxa rather than Teredolites. The manuscript is being in a good shape and deserves publication in PRSB when the authors addressed these shortcomings.

Besides, the manuscript has several critical problems to which the authors need to address before acceptance for publication:

Most serious problems are lying in the systematic account:

1)-1: The formatting of the systematic account is inadequate and it does not fulfill the formalities requested by the ICZN. The description of a species-group taxon (=Lithoredo abatanica) does not include any designation of types (=type specimen series) nor type locality so that the erection of the Lithoredo abatanica as of a species-group taxon violates the ICZN Art. 16.4 (however, as far as I think, two nominal taxon, Lithoredo and Lithoredo abatanica, are deemed to confer availability because the manuscript as a whole could serve as “an action of combined description of a new genus-group taxon and a new species” but the ICZN recommends not to adopt such an action). The authors will remind that a genus-group taxon and a species-group taxon are to be proposed under “different” definition and different Articles of the ICZN. Please accordingly arrange the format into “appropriate” one which fulfills the ICZN. I thus made numbers of corrections and arrangements to overcome this critical problem directly on the manuscript.

1)-2: Designation of type specimens (for *L. abatanica*) are poorly done. The holotype, at least, needs to have the description of the size, locality details, collection date, collector(s), preservation condition, registration number etc. in the TEXT of “Type material” following the ICZN (Recommendation 73C in particular). All the figured and/or analyzed specimens, in my opinion, is better to have a status of paratype. Specimens mentioned in the figs 3A and 3D-I are better designated as paratypes. It is also a good practice to have type status for DNA voucher specimen(s), and ideally, best selected as the holotype following the trend.

1)-3: Please explain the difference of “specimen #” and “accession #”. In general, specimen number is identical with registration (=accession) numbers in taxonomic works. I strongly recommend the authors to use “registration(=accession) number” for all material presented.

1)-4: Irrespective of the fact that the types are still in ANSP, I strongly recommend the authors to ask National Museum of the Philippines (NMP) to issue registration numbers, and then put the registration numbers of the NMP into the manuscript. Otherwise, the authors are subject to publish an addenda when the types are moved into NMP.

Another critical issue is:

2) The authors totally ignored a “most referable” teredinid species. As I told to the first author several years ago, *Lithoredo abatanica* has to be compared with *Teredo ancilla* Barnard, 1964 described from Umlalazi estuary in Zululand. A single specimen collected from a log in mangrove swamp has been known to date but the original description of *T. ancilla* (Barnard, 1964: 568–570, fig. 36) said that this species is “..median part consisting of the anterior area only; ends of the ridges on the posterior margin forming an irregular series of small lobules....No middle area with groove...(p. 568); Pallet club-shaped...club broader than length., externally slightly hollowed, internally slightly biconcave with feeble median rib, distal margin with slight median projection; compared with most species of *Teredo* this specimen has a short and stout body....(p. 569); the valves are remarkable for the reduction in width of the median part, involving the suppression of the middle area with its usual groove (p. 570)”. The description as well as its hand-drawing figures well match with morphology of *Lithoredo abatanica*. Although Turner (1966: 87) has synonymized *T. ancilla* with *Dicyathifer manni* without any explanation, I believe her allocation was going too far and *T. ancilla* should be treated as an independent species – the hand drawings in the original description are good enough to let us conclude this. *Teredo ancilla* appears to be not conspecific with *L. abatanica*, however, the authors have to make a close scrutiny upon the type specimen of *T. ancilla*, other than genera mentioned.

3) Although the authors discussed ichnotaxa/ichnofacies, I feel that they erroneously interpret the concept of ichnotaxa/ichnofacies. Ichnotaxon as well as ichnofacies are defined based solely on the type of substrata and morphologies, and never affected by the types (or biological species) of tracemakers. *Teredolites* is able to use only for trace fossils in woodground. Therefore, traces produced by *Lithoredo* cannot be referable with *Teredolites*: it should be *Trypanites* or *Gastrochaenonites* in rockground or *Glossifungites* in firmground. The authors thus need to review *Trypanites*, *Gastrochaenolites* and *Glossifungites* (and may be *Entobia*) if they wish to give ichnological implications. Bromley (1972), Kelly & Bromley (1984) and MacEachern et al. (2007) are one of the good references to be cited.

Following suggestions and recommendations will be addressed by the authors when they revise the manuscript:

4) The authors are encouraged to prepare a full description of a new species (but now erroneously given under a new genus). The authors may worry about the page limit of the manuscript but descriptions for labial palp, internal structure such as apophysis and condyles, and apertural features of the calcareous tube are highly needed in the main text, even some of them are given only in the Supplementary Table 1. It is also very important to take ontogenetic variations into account. For example, the authors said that umbonal-ventral sulcus (uvs) is absent but this is overlooking: please remind that uvs as well as posterior slope clearly present only in individuals in young stage.

5) This work contains taxonomic account so that I strongly recommend the author to give the author(s) and year in the scientific names which first appears in the text, e.g., *Kuphus polythalamius* (Linnaeus, 1767) at the first appearance.

6) The type of the host rock of *Lithoredo abatanica* is best presented as “weakly consolidated marly packstone” or “marly sediments”, not as limestone.

7) The limited distribution of marly packstone as of the habitat of *L. abatanica* was given only in the final paragraph of the Conclusion from a conservation point of view. The authors clearly articulate that the rock harboring *L. abatanica* exposes in a narrow area in the Abatan River in earlier sections.

8) The authors concluded *L. abatanica* dwells in freshwater condition based on their salinity measurement (<0.5 ppt). However, Lozouet & Plaziat (2008: 11, 92; the survey was conducted May–July, a wet season) has reported the salinity was 2 ppt, suggesting blackish water environment. Bouchet et al. (2009) also listed physiological setting of stations of the PANGLAO 2004 and showed the salinity of the Stn M39 was 2 ‰. My measurement made on June 6, 2007 during low tide (water sampled from near the bottom, ca. 2 meters deep; already wet season) counted 1 ppt. Lozouet & Plaziat (2008) has regarded that *L. abatanica* is a member of estuarine malacofauna, not of freshwater. Considering those observations and river settings of the Abatan River, I think it is safe to treat that *L. abatanica* is an oligohaline–freshwater dweller.

9) In Materials & Methods, the purpose of SEM, Micro CT and X-ray analyses was not clearly articulated so that it is rather hard to understand what analysis contributed to what data. Please briefly explain the reason why the authors implemented those analyses. It may be a good idea to present it at the end of the Introduction.

10) Please rearrange the format of the heading that needs numerical numbering and references suitable for PRSB.

11) Supplementary Figure 2 can be combined with Figure 3. Photographs in Supplementary Figure 2 are indispensable as of the "main" part of the paper.

Please find an annotated manuscript in pdf in which I have made numbers of comments and suggestions. I might be harsh in my review, but hopefully my comments and suggestions will help the authors to improve the manuscript and get it published in PRSB.

References mentioned both in this comments and in the annotated manuscript are:

Betcher, M. A. et al. 2012. Microbial distribution and abundance in the digestive system of five shipworm species (*Bivalvia*: *Teredinidae*). PLoS ONE, 7(9): e45309

Bøggild, O. B. 1930. The shell structure of the mollusks. Det Kongelige Danske Videnskabernes Selskabs Skrifter. Naturvidenskabelig og Matematisk Afdeling, 9: 231–326, 14 pls.

Bromley, R. G. 1972. On some ichnotaxa in hard substrates, with a redefinition of *Trypanites* Mägdefrau. Paläontologische Zeitschrift, 46(1/2): 93–98.

Bouchet, P., Ng, P. K. L., Largo, D. & Tan, S. H. 2009. Panglao 2004 – Investigations of the marine species richness in the Philippines. The Raffles Bulletin of Zoology, Supplement, 20: 1–19.

Carter, J. G. 1980. Selected mineralogical data for the *Bivalvia*. p. 627–643. In, Roads, D. C. & Lutz, R. A. (eds), Skeletal Growth of Aquatic Organisms. Plenum Press, New York & London.

Kelly, S. R. A. & Bromley, R. G. 1984. Ichnological nomenclature of clavate borings. Palaeontology, 27(4): 793–807

MacEachern, J. A., Pemberton, S. G., Gingras, M. K. & Bann, K. L. 2007. The ichnofacies paradigm: A fifty-year retrospective, pp. 52-77. In, Miller III, W. (ed.) Trace fossils concepts, problems, prospects. Elsevier, Netherlands.

Plint, A. G. & Pickerill, R. K. 1985. Non-marine Teredolites from the middle Eocene of southern England. *Lethaia*, 18: 341-347.

Porth, H. 1989. On the petroleum prospects of the Visayan Basin, Philippines. *Geologische Jahrbuch B*, 70: 385-406.

Review form: Reviewer 3 (Luisa M S Borges)

Recommendation

Accept with minor revision (please list in comments)

Scientific importance: Is the manuscript an original and important contribution to its field?

Excellent

General interest: Is the paper of sufficient general interest?

Excellent

Quality of the paper: Is the overall quality of the paper suitable?

Excellent

Is the length of the paper justified?

Yes

Should the paper be seen by a specialist statistical reviewer?

No

Do you have any concerns about statistical analyses in this paper? If so, please specify them explicitly in your report.

No

It is a condition of publication that authors make their supporting data, code and materials available - either as supplementary material or hosted in an external repository. Please rate, if applicable, the supporting data on the following criteria.

Is it accessible?

Yes

Is it clear?

Yes

Is it adequate?

Yes

Do you have any ethical concerns with this paper?

No

Comments to the Author

Dear Colleagues,

Your article describing this highly unusual teredinid species is very interesting and sets the stage nicely for future research work on this teredinid species. The use of several lines of evidence to demonstrate that these organisms should be placed in a new genus and species is very adequate. In addition, the ms is clearly written and can be understood even by non-specialists.

Below are some minor comments and/or suggestions:

Line 55- instead of "all of part" should be "all or part".

Line 270- You say that *Lithoredo* can be distinguished from *Dicyathifer* and *Kuphus* (both with pallets with medial ridge) by the absence of a medial ridge. It seems to me from figure 3 that the inner side of the pallets of *Lithoredo* have a medial ridge, although it is not easy to ascertain this only from the figure. In addition, in Supplemental Table 1 it says that the pallets of *Lithoredo* are medially-divided. Please check this, it seems incongruent to me. In addition, I suggest that a series of different size pallets should be shown. This would show the intraspecific variation in the morphology of the pallets, important for future identification of specimens by other researchers. Line 557- In figure 4, I think "si" is not defined. Please check.

Best wishes

Decision letter (RSPB-2019-0434.R0)

11-Apr-2019

Dear Dr Shipway:

Your manuscript has now been peer reviewed and the reviews have been assessed by an Associate Editor. The reviewers' comments (not including confidential comments to the Editor) and the comments from the Associate Editor are included at the end of this email for your reference. As you will see, the reviewers and the Editors have raised some concerns with your manuscript and we would like to invite you to revise your manuscript to address them.

When submitting your revision please upload a file under "Response to Referees" - in the "File Upload" section. This should document, point by point, how you have responded to the reviewers' and Editors' comments, and the adjustments you have made to the manuscript. We

require a copy of the manuscript with revisions made since the previous version marked as 'tracked changes' to be included in the 'response to referees' document.

Research ethics:

Use of animals and field studies:

If you wish to submit your data to Dryad (<http://datadryad.org/>) and have not already done so you can submit your data via this link [http://datadryad.org/submit?journalID=RSPB&manu=\(Document not available\)](http://datadryad.org/submit?journalID=RSPB&manu=(Document%20not%20available)), which will take you to your unique entry in the Dryad repository.

Online supplementary material will also carry the title and description provided during submission, so please ensure these are accurate and informative. Note that the Royal Society will

not edit or typeset supplementary material and it will be hosted as provided. Please ensure that the supplementary material includes the paper details (authors, title, journal name, article DOI). Your article DOI will be 10.1098/rspb.[paper ID in form xxxx.xxxx e.g. 10.1098/rspb.2016.0049].

Please submit a copy of your revised paper within three weeks. If we do not hear from you within this time your manuscript will be rejected. If you are unable to meet this deadline please let us know as soon as possible, as we may be able to grant a short extension.

Best wishes,

Proceedings B

Board Member: 2

Comments to Author:

All three reviewers raise the same issues of technical shortcomings in the systematics aspect of the paper. These are extremely important and the paper cannot be considered for publication unless these fundamental errors are thoroughly corrected.

The second reviewer also raises a number of points that would improve the scientific context and ensure a broad audience of readers. Most important perhaps is the critical evidence that this is not an obligate freshwater species but instead occurs in estuarine / brackish environments. I would expect the authors to include this in the revision of the work.

Reviewer(s)' Comments to Author:

Referee: 1

Comments to the Author(s)

Dear colleagues,

Oh my God, Fantastic Beasts and Where to Find Them? In the Philippines, of course. A rock-boring and rock-ingesting freshwater shipworm! My God! It looks like an alien creature, not an animal from our planet. Congratulations with this amazing finding! After our discovery of a freshwater rock-boring mussel in Myanmar I didn't think that I could be surprised by anything but now ... now my brain felt as if it was gonna explode.

This manuscript is very well written and concise. As far as I know, it reports one of the greatest discoveries in freshwater malacology and hydrobiology during the last decades, if not the last hundred years. In my opinion, it is fully appropriate for the journal, and could be published after a minor revision.

I have only a few comments as follows:

219: ...using PDXL software Figure 5D => ...using PDXL software (Fig. 5D)

226-257: In a description of the genus-level taxon, you should indicate the type species (i.e. line 225) and provide a brief diagnosis of the genus (i.e. lines 238-243), etymology (lines 244-245), and remarks (lines 258-281). However, the type material (lines 226-231), type locality (line 232), comparative material (lines 233-237), habitat (line 246), and description (lines 247-257) seem to be inappropriate for a description of the genus-level taxon. These paragraphs must be transferred to

the species description. Your abatanica will be a nomen nudum if this description will be published in its current form, because your description does not conform the rules of the ICZN: no reference to the type series is provided; the type locality is not indicated; a description is too short and incomplete.

228-231: "Paratypes are deposited at ANSP, at the Marine Science Institute of the University of the Philippines (MSI) and at the Ocean Genome Legacy Center of New England Biolabs, Northeastern University (OGL)." => This statement does not correspond to Table 1. Please list all available paratypes with their voucher codes in this table.

296: from a fresh-water... => from a freshwater...

283: *Lithoredo abatanica* Shipway & Distel, sp. nov. => *Lithoredo abatanica* Shipway & Distel, gen. & sp. nov.

549: Figure 3: The Morphology... => Figure 3: Morphology...

557: Figure 4: The anatomy... => Figure 4: Anatomy...

596-597: Why you want to transfer the whole type series (lines 596-597) or at least the holotype (lines 226-228) from the ANSP to a Philippine museum? I strongly disagree with this idea. From a historical perspective, there are only few countries and few large museums in the world, in which the long-term storage of type collections can be secured, i.e. the United States (ANSP, MCZ, USNM, and others), UK (BMNH), Germany (SMF, ZMB), France (MNHN), Australia (AMS), and some others. For example, I spent a lot of time transferring the holotypes of my new taxa to the US (e.g. to the North Carolina Museum of Natural Sciences), because I am not sure that they will be safe (in a long-term perspective!) in Russian depositories. Additionally, the types will be much more accessible for researchers in the ANSP than in a Philippine museum. Could you deposit the holotype and 2-3 paratypes in Philadelphia? Several paratypes can also be transferred to the National Museum of the Philippines and other depositories. I guess it would be the best solution. Supp. Fig. 1: clam = Mytilidae (freshwater lineage of *Xenostrobus* sp.?); gastropod = Neretidae (*Septaria* cf. *luzonica*?). Nice animals. Just for information.

I look forward to read this great paper published!

With kind greetings from frozen Arctic Russia,
Ivan Bolotov

Referee: 2

Comments to the Author(s)

I apologize for the delay in reviewing the manuscript.

The manuscript describes a novel finding of an eccentric shipworm from "unusual habitat" and tries to discuss its potential impact. While I appreciate the effort of the work and the significance of this description, I think the authors need to improve the resulting discussion suitable for PRSB with providing more concrete data and scenarios those may attract the readers: implications for rock-ingestion and feeding mode still need more data to clarify their nutrient origin, and to do so, the authors need to show data on possible symbionts in their gills, even if it is concluded to be barren; implication for teredinid evolution needs some plausible scenarios from an obligate wood-boring into obligate rock-boring mode of life, but now the authors only showed phylogenetic position of the shipworm and demonstrated that the shipworm is an obligate rock-borer; and implication for trace fossils needs a comparison with adequate ichnotaxa rather than *Teredolites*. The manuscript is being in a good shape and deserves publication in PRSB when the authors addressed these shortcomings.

Besides, the manuscript has several critical problems to which the authors need to address before acceptance for publication:

Most serious problems are lying in the systematic account:

1)-1: The formatting of the systematic account is inadequate and it does not fulfill the formalities requested by the ICZN. The description of a species-group taxon (=Lithoredo abatanica) does not include any designation of types (=type specimen series) nor type locality so that the erection of the Lithoredo abatanica as of a species-group taxon violates the ICZN Art. 16.4 (however, as far as I think, two nominal taxon, Lithoredo and Lithoredo abatanica, are deemed to confer availability because the manuscript as a whole could serve as “an action of combined description of a new genus-group taxon and a new species” but the ICZN recommends not to adopt such an action). The authors will remind that a genus-group taxon and a species-group taxon are to be proposed under “different” definition and different Articles of the ICZN. Please accordingly arrange the format into “appropriate” one which fulfills the ICZN. I thus made numbers of corrections and arrangements to overcome this critical problem directly on the manuscript.

1)-2: Designation of type specimens (for *L. abatanica*) are poorly done. The holotype, at least, needs to have the description of the size, locality details, collection date, collector(s), preservation condition, registration number etc. in the TEXT of “Type material” following the ICZN (Recommendation 73C in particular). All the figured and/or analyzed specimens, in my opinion, is better to have a status of paratype. Specimens mentioned in the figs 3A and 3D–I are better designated as paratypes. It is also a good practice to have type status for DNA voucher specimen(s), and ideally, best selected as the holotype following the trend.

1)-3: Please explain the difference of “specimen #” and “accession #”. In general, specimen number is identical with registration (=accession) numbers in taxonomic works. I strongly recommend the authors to use “registration(=accession) number” for all material presented.

1)-4: Irrespective of the fact that the types are still in ANSP, I strongly recommend the authors to ask National Museum of the Philippines (NMP) to issue registration numbers, and then put the registration numbers of the NMP into the manuscript. Otherwise, the authors are subject to publish an addenda when the types are moved into NMP.

Another critical issue is:

2) The authors totally ignored a “most referable” teredinid species. As I told to the first author several years ago, *Lithoredo abatanica* has to be compared with *Teredo ancilla* Barnard, 1964 described from Umlalazi estuary in Zululand. A single specimen collected from a log in mangrove swamp has been known to date but the original description of *T. ancilla* (Barnard, 1964: 568–570, fig. 36) said that this species is “..median part consisting of the anterior area only; ends of the ridges on the posterior margin forming an irregular series of small lobules....No middle area with groove...(p. 568); Pallet club-shaped...club broader than length..., externally slightly hollowed, internally slightly biconcave with feeble median rib, distal margin with slight median projection; compared with most species of *Teredo* this specimen has a short and stout body....(p. 569); the valves are remarkable for the reduction in width of the median part, involving the suppression of the middle area with its usual groove (p. 570)”. The description as well as its hand-drawing figures well match with morphology of *Lithoredo abatanica*. Although Turner (1966: 87) has synonymized *T. ancilla* with *Dicyathifer manni* without any explanation, I believe her allocation was going too far and *T. ancilla* should be treated as an independent species – the hand drawings in the original description are good enough to let us conclude this. *Teredo ancilla* appears to be not conspecific with *L. abatanica*, however, the authors have to make a close scrutiny upon the type specimen of *T. ancilla*, other than genera mentioned.

3) Although the authors discussed ichnotaxa/ichnofacies, I feel that they erroneously interpret the concept of ichnotaxa/ichnofacies. Ichnotaxon as well as ichnofacies are defined based solely on the type of substrata and morphologies, and never affected by the types (or biological species) of tracemakers. Teredolites is able to use only for trace fossils in woodground. Therefore, traces produced by *Lithoredo* cannot be referable with *Teredolites*: it should be *Trypanites* or *Gastrochaenolites* in rockground or *Glossifungites* in firmground. The authors thus need to review *Trypanites*, *Gastrochaenolites* and *Glossifungites* (and may be *Entobia*) if they wish to

give ichnological implications. Bromley (1972), Kelly & Bromley (1984) and MacEachern et al. (2007) are one of the good references to be cited.

Following suggestions and recommendations will be addressed by the authors when they revise the manuscript:

4) The authors are encouraged to prepare a full description of a new species (but now erroneously given under a new genus). The authors may worry about the page limit of the manuscript but descriptions for labial palp, internal structure such as apophysis and condyles, and apertural features of the calcareous tube are highly needed in the main text, even some of them are given only in the Supplementary Table 1. It is also very important to take ontogenetic variations into account. For example, the authors said that umbonal-ventral sulcus (uvs) is absent but this is overlooking: please remind that uvs as well as posterior slope clearly present only in individuals in young stage.

5) This work contains taxonomic account so that I strongly recommend the author to give the author(s) and year in the scientific names which first appears in the text, e.g., *Kuphus polythalamius* (Linnaeus, 1767) at the first appearance.

6) The type of the host rock of *Lithoredo abatanica* is best presented as “weakly consolidated marly packstone” or “marly sediments”, not as limestone.

7) The limited distribution of marly packstone as of the habitat of *L. abatanica* was given only in the final paragraph of the Conclusion from a conservation point of view. The authors clearly articulate that the rock harboring *L. abatanica* exposes in a narrow area in the Abatan River in earlier sections.

8) The authors concluded *L. abatanica* dwells in freshwater condition based on their salinity measurement (<0.5 ppt). However, Lozouet & Plaziat (2008: 11, 92; the survey was conducted May-July, a wet season) has reported the salinity was 2 ppt, suggesting blackish water environment. Bouchet et al. (2009) also listed physiological setting of stations of the PANGLAO 2004 and showed the salinity of the Stn M39 was 2 ‰. My measurement made on June 6, 2007 during low tide (water sampled from near the bottom, ca. 2 meters deep; already wet season) counted 1 ppt. Lozouet & Plaziat (2008) has regarded that *L. abatanica* is a member of estuarine malacofauna, not of freshwater. Considering those observations and river settings of the Abatan River, I think it is safe to treat that *L. abatanica* is an oligohaline-freshwater dweller.

9) In Materials & Methods, the purpose of SEM, Micro CT and X-ray analyses was not clearly articulated so that it is rather hard to understand what analysis contributed to what data. Please briefly explain the reason why the authors implemented those analyses. It may be a good idea to present it at the end of the Introduction.

10) Please rearrange the format of the heading that needs numerical numbering and references suitable for PRSB.

11) Supplementary Figure 2 can be combined with Figure 3. Photographs in Supplementary Figure 2 are indispensable as of the "main" part of the paper.

Please find an annotated manuscript in pdf in which I have made numbers of comments and suggestions. I might be harsh in my review, but hopefully my comments and suggestions will help the authors to improve the manuscript and get it published in PRSB.

References mentioned both in this comments and in the annotated manuscript are:

Betcher, M. A. et al. 2012. Microbial distribution and abundance in the digestive system of five shipworm species (Bivalvia: Teredinidae). PLoS ONE, 7(9): e45309

Bøggild, O. B. 1930. The shell structure of the mollusks. Det Kongelige Danske Videnskabernes Selskabs Skrifter. Naturvidenskabelig og Matematisk Afdeling, 9: 231-326, 14 pls.

Bromley, R. G. 1972. On some ichnotaxa in hard substrates, with a redefinition of Trypanites Mägdefrau. Paläontologische Zeitschrift, 46(1/2): 93-98.

Bouchet, P., Ng, P. K. L., Largo, D. & Tan, S. H. 2009. Panglao 2004 – Investigations of the marine species richness in the Philippines. The Raffles Bulletin of Zoology, Supplement, 20: 1-19.

Carter, J. G. 1980. Selected mineralogical data for the Bivalvia. p. 627-643. In, Roads, D. C. & Lutz, R. A. (eds), Skeletal Growth of Aquatic Organisms. Plenum Press, New York & London.

Kelly, S. R. A. & Bromley, R. G. 1984. Ichnological nomenclature of clavate borings. Palaeotology, 27(4): 793-807

MacEachern, J. A., Pemberton, S. G., Gingras, M. K. & Bann, K. L. 2007. The ichnofacies paradigm: A fifty-year retrospective, pp. 52-77. In, Miller III, W. (ed.) Trace fossils concepts, problems, prospects. Elsevier, Netherlands.

Plint, A. G. & Pickerill, R. K. 1985. Non-marine Teredolites from the middle Eocene of southern England. Lethaia, 18: 341-347.

Porth, H. 1989. On the petroleum prospects of the Visayan Basin, Philippines. Geologische Jahrbuch B, 70: 385-406.

Referee: 3

Comments to the Author(s)

Dear Colleagues,

Your article describing this highly unusual teredinid species is very interesting and sets the stage nicely for future research work on this teredinid species. The use of several lines of evidence to demonstrate that these organisms should be placed in a new genus and species is very adequate. In addition, the ms is clearly written and can be understood even by non-specialists.

Below are some minor comments and/or suggestions:

Line 55- instead of "all of part" should be "all or part".

Line 270- You say that *Lithoredo* can be distinguished from *Dicyathifer* and *Kuphus* (both with pallets with medial ridge) by the absence of a medial ridge. It seems to me from figure 3 that the inner side of the pallets of *Lithoredo* have a medial ridge, although it is not easy to ascertain this only from the figure. In addition, in Supplemental Table 1 it says that the pallets of *Lithoredo* are medially-divided. Please check this, it seems incongruent to me. In addition, I suggest that a series of different size pallets should be shown. This would show the intraspecific variation in the morphology of the pallets, important for future identification of specimens by other researchers. Line 557- In figure 4, I think "si" is not defined. Please check.

Best wishes

Author's Response to Decision Letter for (RSPB-2019-0434.R0)

See Appendix A.

RSPB-2019-0434.R1 (Revision)

Review form: Reviewer 1 (Ivan Bolotov)

Recommendation

Accept as is

Scientific importance: Is the manuscript an original and important contribution to its field?

Excellent

General interest: Is the paper of sufficient general interest?

Excellent

Quality of the paper: Is the overall quality of the paper suitable?

Excellent

Is the length of the paper justified?

Yes

Should the paper be seen by a specialist statistical reviewer?

No

Do you have any concerns about statistical analyses in this paper? If so, please specify them explicitly in your report.

No

It is a condition of publication that authors make their supporting data, code and materials available - either as supplementary material or hosted in an external repository. Please rate, if applicable, the supporting data on the following criteria.

Is it accessible?

Yes

Is it clear?

Yes

Is it adequate?

Yes

Do you have any ethical concerns with this paper?

No

Comments to the Author

Dear authors,
Good work. I look forward to reading it in a Proceedings B issue.
Best wishes,
Ivan Bolotov

Review form: Reviewer 2 (Takuma Haga)**Recommendation**

Accept with minor revision (please list in comments)

Scientific importance: Is the manuscript an original and important contribution to its field?

Excellent

General interest: Is the paper of sufficient general interest?

Good

Quality of the paper: Is the overall quality of the paper suitable?

Acceptable

Is the length of the paper justified?

Yes

Should the paper be seen by a specialist statistical reviewer?

No

Do you have any concerns about statistical analyses in this paper? If so, please specify them explicitly in your report.

No

It is a condition of publication that authors make their supporting data, code and materials available - either as supplementary material or hosted in an external repository. Please rate, if applicable, the supporting data on the following criteria.

Is it accessible?

Yes

Is it clear?

Yes

Is it adequate?

Yes

Do you have any ethical concerns with this paper?

No

Comments to the Author

Comments to the authors:

I guess that the revision of the manuscript was a bit of hard work... Thanks to the authors' great effort, the manuscript is now in good shape with convincing data presented in the good

manuscript narrative. I am confident that this work is a very important novel contribution to our knowledge of molluscan evolution, paleoecology, and also a wide field in biology that certainly deserves publication in the PRSB. It's a super work!

This manuscript should therefore be published though, it does need a bit of minor modification (in the systematic account in particular) as mentioned below:

=====

line 68, "Spathoteredo sp.": "sp." should be in upright type.

lines 213–214: It makes sense! Please modify the text of the line 214 as "...sequences data from only one specimen (A34423)....".

line 297: "place" should be "placed".

line 333: "Lithoredo Shipway, Distel & Rosenberg, gen. nov." is better given as "Genus Lithoredo Shipway, Distel & Rosenberg gen. nov.". Better to include "Genus" at the beginning!

line 380: Please insert a phrase that "The Genus Lithoredo is monotypic at present" at the beginning .

line 387: "...gen. nov. may be distinguished from...." should be "...gen. nov. is distinguished from...".

line 487: Please delete a word "Synonymy". This record by Lozouet & Plaziat (2008) is not of a nomenclatural synonym – just a "list of illustrated/figured record".

lines 494–496: The authors say that they've modified the designation of the type series but I was not able to find any change neither in the text nor Table 1. In the revised text the authors say the paratypes are also housed in the Ocean Genome Legacy Center but none of additional designation of the paratype was made in the Table 1. Please rectify this. In general, as of the formality, we give full specimen details even for paratypes. My practical suggestion is that the authors modify the texts: "Type material: Holotype, ANSP A477140, measuring 105.4 mm in total body length, fixed in 4 % formaldehyde and stored in 70 % ethanol; Paratype 1, ANSP A477141, "number of specimens, other details"; Paratype 2, ANSP A477141, "number of specimens, other details"; Paratype..... The holotype and paratype specimens were collected from the type locality on Aug. 20, 2018 by JRS, MAA and Melfeb Chicote." Please remind that the prefix "PMS-####" is NOT of the formal repository (just a handling number of the authors/project) so that the authors should adopt the prefix starting with "ANSP" or those given by the Ocean Genome Legacy.

lines 498–502: The authors should move a whole sentence into "Materials & Methods". Specimens and genera listed therein are unnecessary in the systematic account. Please create a section "other material examined" instead and insert a text for example "other specimens of *Lithoredo abatanica* studied are stored in XXXX (YYYY–ZZZZ [accession numbers])". Because the authors did not give any details of figured specimens that do not hold type status (i.e., figured specimens in fig. 3D–I, supplemental fig. 2D–E), researchers cannot access to them.

lines 565–566: "...the illustration of *Teredo ancilla* in Barnard (1964) (15), but...." should be "...the illustration of *Teredo ancilla* Barnard, 1964 (15), but....".

line 593, "Several taxonomic characters identify *L. abatanica* as a new genus and species": This sentence sound strange because identification is, in my opinion, of human work. As I commented in an earlier version of the manuscript, this text is better written for example "Several taxonomic characters led us to establish *L. abatanica* as a new genus and species".

line 659 and line 700: "...the rock-boring mussel Lignopholas...." should be "...the rock-boring piddock Lignopholas....". The pholadid Lignopholas is not a mussel. Please adopt a commonly used word "piddock" which denotes pholadid bivalves.

=====

I am sure that the authors can easily modify these minor modifications and look forward to seeing this impressive contribution published!

Decision letter (RSPB-2019-0434.R1)

29-May-2019

Dear Dr Shipway

I am pleased to inform you that your manuscript entitled "A rock-boring and rock-ingesting freshwater bivalve (shipworm) from the Philippines" has been accepted for publication in Proceedings B.

Open Access

Paper charges

All supplementary materials accompanying an accepted article will be treated as in their final form. They will be published alongside the paper on the journal website and posted on the online

figshare repository. Files on figshare will be made available approximately one week before the accompanying article so that the supplementary material can be attributed a unique DOI.

Sincerely,

Proceedings B
mailto: proceedingsb@royalsociety.org

Associate Editor:
Board Member: 1
Comments to Author:
(There are no comments.)

Board Member: 2
Comments to Author:
(There are no comments.)

Appendix A

30^h April 2019

Dear Editor(s)

Thank you for your comments and suggestions, and those of the reviewers, for the manuscript titled '*A rock-boring and rock-ingesting freshwater bivalve (shipworm) from the Philippines*' (RSPB-2019-0434).

After careful consideration, we have made several changes to the manuscript, outlined below. We feel that the suggestions of the reviewers have greatly improved the quality of this manuscript.

Thank you once again for your comments, we hope you will find the updated manuscript adequately addresses the reviewer's suggestions and we look forward seeing the final publication.

Kind regards

J. Reuben Shipway

Editors Comments

Board Member Comments

Board Member 2 [BMII - a]: All three reviewers raise the same issues of technical shortcomings in the systematics aspect of the paper. These are extremely important and the paper cannot be considered for publication unless these fundamental errors are thoroughly corrected.

Author (BMII - a): We have made all the necessary changes to the systematics in the paper and wish to thank you and the three reviewers for these insightful critiques.

Board Member 2 [BMII - b]: The second reviewer also raises a number of points that would improve the scientific context and ensure a broad audience of readers. Most important perhaps is the critical evidence that this is not an obligate freshwater species but instead occurs in estuarine / brackish environments. I would expect the authors to include this in the revision of the work.

Author (BMII - b): Following both the reviewer and board members recommendations, we have added data on the symbiosis in *Lithoredo* (Supplemental Figure 5, Line 702; Discussion, line 367). In addition, we have provided further details on the salinity measurements at our collection site, including an updated map (Figure 1, Line 554), which highlights the Lozouet and Plaziat (2008) collection site.

Reviewer 1 (Ivan Bolotov) Comments

Comments for the author

Reviewer 1a: Oh my God, Fantastic Beasts and Where to Find Them? In the Philippines, of course. A rock-boring and rock-ingesting freshwater shipworm! My God! It looks like an alien creature, not an animal from our planet. Congratulations with this amazing finding! After our discovery of a freshwater rock-boring mussel in Myanmar I didn't think that I could be surprised by anything but now ... now my brain felt as if it was gonna explode. This manuscript is very well written and concise. As far as I know, it reports one of the greatest discoveries in freshwater malacology and hydrobiology during the last decades, if not the last hundred years. In my opinion, it is fully appropriate for the journal, and could be published after a minor revision.

Author (1a): We sincerely thank Dr Bolotov for his enthusiasm for our research.

Reviewer 1b: 219: ...using PDXL software Figure 5D => ...using PDXL software (Fig. 5D)

Author (1b): changed as requested.

Reviewer 1c: 226-257: In a description of the genus-level taxon, you should indicate the type species (i.e. line 225) and provide a brief diagnosis of the genus (i.e. lines 238-243), etymology (lines 244-245), and remarks (lines 258-281). However, the type material (lines 226-231), type locality (line 232), comparative material (lines 233-237), habitat (line 246), and description (lines 247-257) seem to be inappropriate for a description of the genus-level taxon. These paragraphs must be transferred to the species description. Your *abatanica* will be a nomen nudum if this description will be published in its current form, because your description does not conform the rules of the ICZN: no reference to the type series is provided; the type locality is not indicated; a description is too short and incomplete.

Author (1c): changed as requested.

Reviewer 1d: 228-231: “Paratypes are deposited at ANSP, at the Marine Science Institute of the University of the Philippines (MSI) and at the Ocean Genome Legacy Center of New England Biolabs, Northeastern University (OGL).” => This statement does not correspond to Table 1. Please list all available paratypes with their voucher codes in this table.

Author (1d): changed as requested.

Reviewer 1e: 296: from a fresh-water... => from a freshwater...

Author (1e): changed as requested.

Reviewer 1f: 283: *Lithoredo abatanica* Shipway & Distel, sp. nov. => *Lithoredo abatanica* Shipway & Distel, gen. & sp. nov.

Author (1f): changed as requested.

Reviewer 1g: 549: Figure 3: The Morphology... => Figure 3: Morphology...

Author (1g): changed as requested.

Reviewer 1h: 557: Figure 4: The anatomy... => Figure 4: Anatomy...

Author (1h): changed as requested.

Reviewer 1i: 596-597: Why you want to transfer the whole type series (lines 596-597) or at least the holotype (lines 226-228) from the ANSP to a Philippine museum? I strongly disagree with this idea. From a historical perspective, there are only few countries and few large museums in the world, in which the long-term storage of type collections can be secured, i.e. the United States (ANSP, MCZ, USNM, and others), UK (BMNH), Germany (SMF, ZMB), France (MNHN), Australia (AMS), and some others. For example, I spent a lot of time transferring the holotypes of my new taxa to the US (e.g. to the North Carolina Museum of Natural Sciences), because I am not sure that they will be safe (in a long-term perspective!) in Russian depositories. Additionally, the types will be much more accessible for researchers in the ANSP than in a Philippine museum. Could you deposit the holotype and 2-3 paratypes in Philadelphia? Several paratypes can also be transferred to the National Museum of the Philippines and other depositories. I guess it would be the best solution.

Author (1i): Our collection permit from the Philippine Bureau of Fisheries and Aquatic Resources requires that the holotype is ultimately deposited to the National Museum of the Philippines (NMP). However, the NMP is not currently accepting specimens due to ongoing organisational restructuring. For this reason, we have obtained permission to deposit the holotype to the ANSP until it can be accepted by the NMP. Please note, additional type material will remain at ANSP and Ocean Genome Legacy after the holotype is transferred.

Reviewer 1j: Supp. Fig. 1: clam = Mytilidae (freshwater lineage of *Xenostrobus* sp.?); gastropod = Neretidae (*Septaria* cf. *luzonica*?). Nice animals. Just for information.

Author (1j): thank you for the identifications. We have now included this information in the results.

Reviewer 2 (Anonymous) Comments

Comments for the author

I apologize for the delay in reviewing the manuscript.

Reviewer 2a: The manuscript describes a novel finding of an eccentric shipworm from “unusual habitat” and tries to discuss its potential impact. While I appreciate the effort of the work and the significance of this description, I think the authors need to improve the resulting discussion suitable for PRSB with providing more concrete data and scenarios those may attract the readers: implications for rock-ingestion and feeding mode still need more data to clarify their nutrient origin, and to do so, the authors need to show data on possible symbionts

in their gills, even if it is concluded to be barren; implication for teredinid evolution needs some plausible scenarios from an obligate wood-boring into obligate rock-boring mode of life, but now the authors only showed phylogenetic position of the shipworm and demonstrated that the shipworm is an obligate rock-borer; and implication for trace fossils needs a comparison with adequate ichnotaxa rather than Teredolites. The manuscript is being in a good shape and deserves publication in PRSB when the authors addressed these shortcomings.

Author (2a): Reviewer 2 raises an excellent question regarding mode of nutrition and symbiosis in *Lithoredo*. We are actively working on this question. Our data indicates that *Lithoredo* partners with a complex community of bacteria, several of which belong to taxa not previously known to exist as animal symbionts. Given the uniqueness of this symbiosis, an adequate treatment of this topic is beyond the scope of the current manuscript but will be the subject of a separate published work currently in preparation. However, following the reviewer's recommendations, we have modified the text to indicate the presence of gill endosymbionts (see lines 368-375) and have provided a new supplemental figure (Figure S5), a scanning electron micrograph showing the presence of bacteria in the gills of *Lithoredo*. For the editor's convenience, please find this new figure and accompanying legend copied below:

Supplemental Figure 5: Symbiotic bacteria associated with in the gill of *Lithoredo abatanica*. A, scanning electron micrograph depicting bacteria exposed by fracturing the gill along the plane perpendicular to its long axis; B, boxed region from A. Scale bars for A-B = 100 μ m and 20 μ m respectively.

Reviewer 2 states that the ‘**implication for trace fossils needs a comparison with adequate ichnotaxa rather than Teredolites**’. We fully agree and have treated this topic in detail in a

separate manuscript entitled ‘**Shipworm Bioerosion of Lithic Substrates in a Freshwater Setting, Abatan River, Philippines: Ichnologic, Paleoenvironmental and Biogeomorphological Implications**’ [PONE-D-19-06138]. A copy of this manuscript, which is currently under review, has been provided for the editor’s convenience (see attached PDF).

Reviewer 2b: Besides, the manuscript has several critical problems to which the authors need to address before acceptance for publication:

Most serious problems are lying in the systematic account:

1)-1: The formatting of the systematic account is inadequate and it does not fulfill the formalities requested by the ICZN. The description of a species-group taxon (=Lithoredo abatanica) does not include any designation of types (=type specimen series) nor type locality so that the erection of the Lithoredo abatanica as of a species-group taxon violates the ICZN Art. 16.4 (however, as far as I think, two nominal taxon, Lithoredo and Lithoredo abatanica, are deemed to confer availability because the manuscript as a whole could serve as “an action of combined description of a new genus-group taxon and a new species” but the ICZN recommends not to adopt such an action). The authors will remind that a genus-group taxon and a species-group taxon are to be proposed under “different” definition and different Articles of the ICZN. Please accordingly arrange the format into “appropriate” one which fulfills the ICZN. I thus made numbers of corrections and arrangements to overcome this critical problem directly on the manuscript.

Author (2b): changed as requested.

Reviewer 2c: 1)-2: Designation of type specimens (for L. abatanica) are poorly done. The holotype, at least, needs to have the description of the size, locality details, collection date, collector(s), preservation condition, registration number etc. in the TEXT of “Type material” following the ICZN (Recommendation 73C in particular). All the figured and/or analyzed specimens, in my opinion, is better to have a status of paratype. Specimens mentioned in the figs 3A and 3D–I are better designated as paratypes. It is also a good practice to have type status for DNA voucher specimen(s), and ideally, best selected as the holotype following the trend.

Author (2c): The requested modifications have been made with the exception of the designation of specimen PMS-4313H from Figure 3 as a paratype. This specimen was utilized for histological sectioning.

Reviewer 2d: 1)-3: Please explain the difference of “specimen #” and “accession #”. In general, specimen number is identical with registration (=accession) numbers in taxonomic works. I strongly recommend the authors to use “registration(=accession) number” for all material presented.

Author (2d): changed as requested, see lines 81-82.

Reviewer 2e: 1)-4: Irrespective of the fact that the types are still in ANSP, I strongly recommend the authors to ask National Museum of the Philippines (NMP) to issue registration numbers, and then put the registration numbers of the NMP into the manuscript. Otherwise, the authors are subject to publish an addenda when the types are moved into NMP.

Author (2e): The NMP is not able to accept these samples or provide accession numbers at this time (please refer to author comment 1i above). NMP accession numbers will be assigned upon transfer of the holotype and an addendum will be published at that time.

Reviewer 2f: Another critical issue is:

2) The authors totally ignored a “most referable” teredinid species. As I told to the first author several years ago, *Lithoredo abatanica* has to be compared with *Teredo ancilla* Barnard, 1964 described from Umlalazi estuary in Zululand. A single specimen collected from a log in mangrove swamp has been known to date but the original description of *T. ancilla* (Barnard, 1964: 568–570, fig. 36) said that this species is “..median part consisting of the anterior area only; ends of the ridges on the posterior margin forming an irregular series of small lobules....No middle area with groove...(p. 568); Pallet club-shaped...club broader than length..., externally slightly hollowed, internally slightly biconcave with feeble median rib, distal margin with slight median projection; compared with most species of *Teredo* this specimen has a short and stout body...(p. 569); the valves are remarkable for the reduction in width of the median part, involving the suppression of the middle area with its usual groove (p. 570)”. The description as well as its hand-drawing figures well match with morphology of *Lithoredo abatanica*. Although Turner (1966: 87) has synonymized *T. ancilla* with *Dicyathifer manni* without any explanation, I believe her allocation was going too far and *T. ancilla* should be treated as an independent species—the hand drawings in the original description are good enough to let us conclude this. *Teredo ancilla* appears to be not

conspecific with *L. abatanica*, however, the authors have to make a close scrutiny upon the type specimen of *T. ancilla*, other than genera mentioned.

Author (2f): It is challenging to compare *Lithoredo abatanica* with *Teredo ancilla* (Barnard, 1964). The latter is described based on a single specimen, and only a single hand-drawn figure with brief accompanying text are available for comparison. The manuscript does not report type material.

In the description by Barnard, the shell valves of *T. ancilla* are described as similar to *K. polythalamius* (see below), and the siphons as ‘**separate throughout their length**’, featuring ‘**few inconspicuous**’ papillae. Contrastingly, the siphons of *L. abatanica* are almost entirely united (up to $\frac{3}{4}$ of their entire length) and feature multiple rows of large conspicuous papillae on both the incurrent and excurrent aperture. As siphons are useful taxonomic characters in the Teredinidae (Turner, 1966, Nair and Saraswathy, 1971, Shipway 2016; Shipway 2019), *L. abatanica* is clearly morphologically distinguishable from *T. ancilla*. Additionally, Barnard states that the intestine is visible through the thin mantle on the ventral surface in *T. ancilla*, whereas the ventral mantle in *L. abatanica* is thick and opaque (Figure 4 C-F), and the intestine is only visible dorsally (Figure 3B). This is a major anatomical distinction between the two species.

Barnard notes the similarity of *T. ancilla* to *Kuphus arenaria* (now recognized as *K. polythalamius*). He states that the pallets of *T. ancilla* **resemble** those of *D. manni* and that the shell valves resembled ‘**those of [Teredo] dubia**’, a species which has since been synonymized as *K. polythalamius* (<http://www.marinespecies.org/aphia.php?p=taxdetails&id=138539>). Reviewer 2 notes that Turner (1966) synonymized *T. ancilla* as *Dicyathifer manni* (formerly *Kuphus manni*). The pallets of *D. manni* are nearly indistinguishable from *Kuphus polythalamius*, but the latter species was thought at that time to burrow exclusively in sediments, likely explaining the synonymization. Barnard distinguishes *T. ancilla* from *K. polythalamius* based primarily on the fact that ‘**the animal is a wood-borer**’. However, *K. polythalamius* has since been shown to start life in wood, before transitioning to marine sediments (Shipway 2018). As the morphological characters of the pallets and shell valves of *T. ancilla* are consistent with those of *K. polythalamius*, and *T. ancilla* was reported as a ‘**wood-borer**’ collected from a mangrove log, we suggest that *T. ancilla*, if it is not *D. manni* as proposed by Turner, is most likely *K. polythalamius*. Unfortunately, no type material exists to allow further analysis of the

identity of this single specimen and the few existing drawings provide insufficient detail for any meaningful comparison. Please note that the phylogenetic analysis reported in our manuscript, which includes representatives of both *Dicyathifer* and *Kuphus*, provides robust support for *Lithoredo* as a genus distinct from those to which *T. ancilla* has been compared.

To address the concerns raised by Reviewer 2, a section comparing *L. abatanica* with *T. ancilla* has been added to 'Remarks' in the taxonomic description section (Lines 300-307). For the editor's convenience, this section has been copied below:

'Remarks: *L. abatanica* is superficially similar to the illustration of *Teredo ancilla* in Barnard (1964) (15), but differs in several important ways: 1) the siphons of *T. ancilla* are separate along their entire length and feature few inconspicuous papillae, conversely the siphons of *L. abatanica* are almost entirely united (up to $\frac{3}{4}$ of their entire length) and feature multiple rows of large papillae on both the incurrent and excurrent aperture; 2) the intestine is visible through the thin mantle on the ventral surface in *T. ancilla*, whereas the ventral mantle in *L. abatanica* is thick and opaque, and the intestine is only visible dorsally; 3) *T. ancilla* was found in mangrove wood, not rock.'

Reviewer 2g: 3) Although the authors discussed ichnotaxa/ichnofacies, I feel that they erroneously interpret the concept of ichnotaxa/ichnofacies. Ichnotaxon as well as ichnofacies are defined based solely on the type of substrata and morphologies, and never affected by the types (or biological species) of tracemakers. Teredolites is able to use only for trace fossils in woodground. Therefore, traces produced by *Lithoredo* cannot be referable with Teredolites: it should be Trypanites or Gastrochaenonites in rockground or Glossifungites in firmground. The authors thus need to review Trypanites, Gastrochaenolites and Glossifungites (and may be Entobia) if they wish to give ichnological implications. Bromley (1972), Kelly & Bromley (1984) and MacEachern et al. (2007) are one of the good references to be cited.

Author (2g): We recognize that ichnotaxa identify traces and that traces similar to the burrows of *Lithoredo* in rockground are classified as Gastrochaenolites. Here we simply point out that in cases in which the substrates were ambiguous or unknown, "*Teredolites-like*" casts and/or linings have been interpreted as evidence for the pre-existence of marine woodgrounds and that this may be based on an incorrect assumption. We have discussed the ichnotaxonomic implications of *Lithoredo* in detail in a separate manuscript which is currently under review. Please see the earlier response [Author (2a)] for further details.

Reviewer 2h: Following suggestions and recommendations will be addressed by the authors when they revise the manuscript:

4) The authors are encouraged to prepare a full description of a new species (but now erroneously given under a new genus). The authors may worry about the page limit of the manuscript but descriptions for labial palp, internal structure such as apophysis and condyles, and apertural features of the calcareous tube are highly needed in the main text, even some of them are given only in the Supplementary Table 1. It is also very important to take ontogenetic variations into account. For example, the authors said that umbonal-ventral sulcus (uvs) is absent but this is overlooking: please remind that uvs as well as posterior slope clearly present only in individuals in young stage.

Author (2h): A supplemental figure (Supplemental Figure 2) displaying the ontogenetic variation of calcareous structures (pallets and shell valves) has now been included in the manuscript (Line 657-672). For the editor's convenience, please find this figure displayed below. Additional details on the labial palps, internal structures of the valves and apertural features of the calcareous tube have been added to the description (Lines 287-297).

Supplemental Figure 2. Supplemental Figure 2: Calcareous shell valves of *Lithoredo abatanica*: A, MicroCT 3D render of dorsal animal (specimen PMS-4314K); B, outer face shell valves (specimen PMS-4130P); C, inner face shell valves (specimen PMS-4130P). D and E, inner and outer surface of shell valves and pallets across ontogeny. A, apophysis; AS, anterior slope; DC, dorsal condyle; MS, median slope; PAM, posterior adductor muscle; PB,

pallet blade; PS, posterior slope; Si, siphon; SV, shell valve; VC, ventral condyle. Scale bar for A, B-C and D-E = 2.5 mm, 1 mm and 2 mm respectively.

Reviewer 2i:) This work contains taxonomic account so that I strongly recommend the author to give the author(s) and year in the scientific names which first appears in the text, e.g., *Kuphus polythalamius* (Linnaeus, 1767) at the first appearance.

Author (2i): changed as requested.

Reviewer 2j: 6) The type of the host rock of *Lithoredo abatanica* is best presented as “weakly consolidated marly packstone” or “marly sediments”, not as limestone.

Author (2j): we have characterized the rock fully as limestone in a separate paper which is currently under review. Please see the earlier response [Author (2a)] for further details.

Reviewer 2k: 7) The limited distribution of marly packstone as of the habitat of *L. abatanica* was given only in the final paragraph of the Conclusion from a conservation point of view. The authors clearly articulate that the rock harboring *L. abatanica* exposes in a narrow area in the Abatan River in earlier sections.

Author (2k): We are not certain what the reviewer is requesting here. Our guess is that the reviewer meant to say “The authors (should) clearly articulate that the rock harboring *L. abatanica* exposes in a narrow area in the Abatan River in earlier sections

In response to this we have pointed out the limited geographic range of this species in lines 176-180 and 414-424 and in Figure 1.

Reviewer 2l: 8) The authors concluded *L. abatanica* dwells in freshwater condition based on their salinity measurement (<0.5 ppt). However, Lozouet & Plaziat (2008: 11, 92; the survey was conducted May–July, a wet season) has reported the salinity was 2 ppt, suggesting blackish water environment. Bouchet et al. (2009) also listed physiological setting of stations of the PANGLAO 2004 and showed the salinity of the Stn M39 was 2 %. My measurement made on June 6, 2007 during low tide (water sampled from near the bottom, ca. 2 meters deep; already wet season) counted 1 ppt. Lozouet & Plaziat (2008) has regarded that *L. abatanica* is a member of estuarine malacofauna, not of freshwater. Considering those observations and river settings of the Abatan River, I think it is safe to treat that *L. abatanica* is an oligohaline–freshwater dweller.

Author (2l): Our salinity measurements, collected at both high and low tides, were carried out at a collection site approximately 2km (1.94 km/1.2 mi, as measured on Google Maps) upstream ($9^{\circ}45'56.1''\text{N}$ $123^{\circ}56'39.3''\text{E}$) of Stn M39 ($9^{\circ}45.2'\text{N}$, $123^{\circ}56.0'\text{E}$) from Lozouet and Plaziat (2008) and just downstream of Kawasan Falls which provides a very large input of pure freshwater. This likely accounts for the difference in salinity measurements. Regardless of previous findings reporting *Lithoredo* in slightly more brackish waters, our collection site was at a freshwater site. Therefore, our general comments regarding freshwater bioerosion are still accurate and applicable. As suggested by Reviewer 2 we have provided additional information regarding Plaziat and Lozouet's (2008) (Line 176-182) collection site, updated the 'Habitat' section in the 'Systematics' description to include oligohaline distribution for *Lithoredo* (Line 281) and we discuss the previous authors finding in the 'Discussion' (Line 414-424). Additionally, to highlight the previous work by Lozouet and Plaziat (2008) and demonstrate the geographic distance between the previous collection site and the site reported herein, we have redesigned the map in Figure 1. For the editor's convenience, please find updated Figure 1 below and the corresponding legend:

Figure 1: Specimen collection site. A, Bohol Island, Philippines; B, boxed region from A showing an overview of the Abatan River system; C, collection site location. Yellow pin, Lozouet and Plaziat (2008) station M39 ($9^{\circ}45.2'\text{N}$, $123^{\circ}56.0'\text{E}$); red pin, collection site from this study ($9^{\circ}45'56.1''\text{N}$ $123^{\circ}56'39.3''\text{E}$); blue pin, Kawasan Falls.

Reviewer 2m: 9) In Materials & Methods, the purpose of SEM, Micro CT and X-ray analyses was not clearly articulated so that it is rather hard to understand what analysis contributed to what data. Please briefly explain the reason why the authors implemented those analyses. It may be a good idea to present it at the end of the Introduction.

Author (2m): A sentence briefly explaining the purpose of the SEM, MicroCT and X-ray analysis was been provided at the beginning of each respective subsection in the methodology.

Reviewer 2n: 10) Please rearrange the format of the heading that needs numerical numbering and references suitable for PRSB.

Author (2n): This manuscript is consistent with the journal's policy of format-free initial submission. All necessary formatting changes will be made if/when this paper is accepted for publication.

Reviewer 2o: 11) Supplementary Figure 2 can be combined with Figure 3. Photographs in Supplementary Figure 2 are indispensable as of the "main" part of the paper.

Author (2o): Figure 3 provides the first images of this new genus/species, as well as the pallets and shell valve - key taxonomic characters of the Teredinidae. We feel that this figure is already well suited to introducing general readers (who are likely unfamiliar with

shipworms) to this bizarre animal, as well as providing a good overview for purposes of basic taxonomic identification.

Supplemental Figure 2 (Line 671) provides an overview of specific shell valve features and specialized terminology, with the purpose of providing additional details for taxonomic identification. This figure is likely of limited interest and functionality to the general reader. We worry that by combining these two large figures together, we will diminish the original purposes of both, and reduce the size and visualized detail of key taxonomic characters useful for identification. For these reasons, we prefer not to combine these two figures and instead keep Figure 3 the same.

However, following the earlier comment by Reviewer 2 (2h) and the suggestion from Reviewer 3 (comment 3c), we have produced a new figure showing the ontogenic variation of the pallets and shell valves. This new figure was combined with the original Supplemental Figure 2. Please refer to comment 2h above.

Reviewer 3 (Anonymous) Comments

Reviewer 3a: Dear Colleagues,

Your article describing this highly unusual teredinid species is very interesting and sets the stage nicely for future research work on this teredinid species. The use of several lines of evidence to demonstrate that these organisms should be placed in a new genus and species is very adequate. In addition, the ms is clearly written and can be understood even by non-specialists.

Author (3a): We thank the third anonymous reviewer for their kind words.

Reviewer 3b: Below are some minor comments and/or suggestions:

Line 55- instead of “all of part” should be “all or part”.

Author (3b): changed as requested.

Reviewer 3c: Line 270- You say that *Lithoredo* can be distinguished from *Dicyathifer* and *Kuphus* (both with pallets with medial ridge) by the absence of a medial ridge. It seems to me from figure 3 that the inner side of the pallets of

Lithoredo have a medial ridge, although it is not easy to ascertain this only from the figure. In addition, in Supplemental Table 1 it says that the pallets of *Lithoredo* are medially-divided. Please check this, it seems incongruent to me. In addition, I suggest that a series of different size pallets should be shown. This would show the intraspecific variation in the morphology of the pallets, important for future identification of specimens by other researchers.

Author (3c): we have now produced a supplemental figure showing the ontogenic variation of the calcareous structures, please refer to Author comment 2h above.

Reviewer 3d: Line 557- In figure 4, I think “si” is not defined. Please check.

Author (3d): changed as requested.